# The impact of atmospheric motions on source-specific black carbon and the induced direct radiative effects over a river-valley region

Huikun Liu[1], Qiyuan Wang[1,2,3*], Suixin Liu[1,5], Bianhong Zhou[4], Yao Qu[1], Jie Tian[1], Ting Zhang[1], Yongming Han[1,2,3], Junji Cao[6,1,5*]

[1]Key Laboratory of Aerosol Chemistry and Physics, State Key Laboratory of Loess and Quaternary Geology, Institute of Earth Environment, Chinese Academy of Sciences, Xi'an, 710061, China
[2]CAS Center for Excellence in Quaternary Science and Global Change, Xi'an, 710061, China
[3]Guanzhong Plain Ecological Environment Change and Comprehensive Treatment National Observation and Research Station, Xi'an 710061, China
[4]Shaanxi Key Laboratory of Disaster Monitoring and Mechanism Simulation, College of Geography & Environment, Baoji University of Arts & Sciences, Baoji 721013, China
[5]Shaanxi Key Laboratory of Atmospheric and Haze-fog Pollution Prevention, Xi'an 710061, China
[6]Institute of Atmospheric Physics, Chinese Academy of Sciences, Beijing 100029, China

*Correspondence to*: Qiyuan Wang (wangqy@ieecas.cn) and Junji Cao (jjcao@mail.iap.ac.cn)

**Abstract.** Black carbon (BC) is one of the most important short lived climate forcers, and atmospheric motions play an important role in determining its mass concentrations of pollutants. Here an intensive observation was launched in a typical river-valley city to investigate relationships between atmospheric motions and BC aerosols. Equivalent BC (eBC) source apportionment was based on an aethalometer model with the site-dependent absorption Ångström exponents (AAEs) and the mass absorption cross-sections (MACs) retrieved using a positive matrix factorization (PMF) model based on observed chemical components (i.e., EC, POC, $K^+$, Mg, Al, Si, S, Cl, Ca, V, Mn, Fe, Ni, Cu, As, Se, Br, Sr, Pb, Ga, and Zn) and primary absorption coefficients at selected wavelengths from $\lambda$ = 370 to 880nm. The derived AAEs from 370 to 880nm were 1.07 for diesel vehicular emissions, 2.13 for biomass burning, 1.74 for coal combustion, and 1.78 for mineral dust. The mean values for $eBC_{fossil}$ and $eBC_{biomass}$ were 2.46 μg m$^{-3}$ and 1.17 μg m$^{-3}$ respectively. Wind run distances and the vector displacements of the wind in 24 h were used to construct a self-organizing map, from which four atmospheric motions categories were identified (local-scale dominant, local-scale strong and regional-scale weak, local-scale weak and regional-scale strong and regional-scale dominant). BC pollution was found to be more likely when the influence of local-scale motions outweighed those of regional-scale motions. Cluster analysis for the back trajectories of air mass calculated by Hybrid Single-Particle Lagrangian Integrated Trajectory model at the study site indicated that the directions of air flow can have different impacts for different scales of motion. The direct radiative effects (DRE) of source-specific eBCs were lower when the influence of regional-scale motions outweighed that of the local ones. However, due to chemical aging of the particles during transport—the DRE efficiencies under regional scale motions were ~1.5 times higher than those under more local influences. The finding that the DRE efficiency of BC increased during the regional transport suggested significant consequences in regions downwind of pollution sources and emphasizes the importance of regionally transported BC for potential climatic effects.

## 1 Introduction

Black carbon (BC) is produced by the incomplete combustion of biomass and fossil fuels. The BC aerosol has a strong light absorption capacity and can cause heating of the atmosphere. In fact, BC is widely recognized as one of the most important short-lived climate forcers (IPCC, 2021). Due to this high light-absorbing ability, BC has the potential to perturb the radiative balance between the earth and atmosphere and in so doing cause in the climate to change and drive ecosystems away from their natural states (Schroter et all., 2005). Those changes ultimately will affect biodiversity and could threaten humans' food security (Ochoa-Hueso et al., 2017; Shindell et al., 2012). Besides heating the atmosphere directly, BC also is important for nucleating clouds, and that is another way in which the particles can cause indirect climatic effects (Jacobson, 2002). As BC is heterogeneously distributed in the atmosphere, its climatic effects are highly variable and dependent on its distribution in the atmosphere, both horizontally and vertically; its radiative properties and how they are affected by of chemical processing; and its lifetime (IPCC, 2021).

The radiative efficiency of BC can vary due to differences in emission sources and atmospheric aging processes (Bond et al., 2013; He et al., 2015; Cappa et al., 2012). Indeed, BC from different sources can vary in light absorbing abilities (Cheng et al., 2011) which can affect the radiative forcing of climate. In addition to the effects of the sources, regional transport can impact the light-absorbing ability through chemical processing or aging (Zhang et al., 2019). After BC particles are emitted, they can stay in the atmosphere for days or a few weeks (IPCC, 2021). During transport, fresh BC can experience a series of physical and chemical changes, for instance, mixing with other substances that can alter its microphysical and optical properties (Kahnert and Kanngießer, 2020). The aging processes can be even faster in polluted regions (Peng et al., 2016), and as a result, the light-absorbing ability of BC can be strongly affected. Indeed, the light absorption ability of BC after aging can be as much as 2.4 times that of fresh particles (Peng et al., 2016).

The concentrations of BC are controlled by local emissions and regional transport, but meteorological conditions also are important because they affect both transport and removal. Normally, local emissions in urban areas are predictable to some degree because those emission sources are mainly anthropogenic and the concentrations of pollutants follow the diurnal patterns driven by anthropogenic activities. By contrast, meteorological conditions and regional transport are governed by multiple scales of motion which result in distinct meteorological impacts on ambient pollutant levels (Levy et al., 2010, Dutton, 1976). A commonly accepted classification of the scale of motion is based on horizontal distance and time scales. Typically, the time scale of local-scale motions varies from hours to days and the spatial scale ranges from $10^2$ to $10^5$ m (Oke et al., 2002; Seinfeld and Pandis, 2006). The local scales of motion are mainly controlled by local factors such as the roughness of the earth's surface, orography, land breeze/sea breeze circulation, etc. (Hewitson and Crane, 2006; IPCC, 2021). Larger scale of motions are associated with a mesoscale or synoptic scale weather systems, which on the one hand can transport pollutants but on the other can disperse them (Kalthoff et al., 2000; Zhang et al., 2012).

The relationships between atmospheric motions and pollutant concentrations are complex. Atmospheric motions determine where and how extensive the pollution impacts are, but of course the rates of pollutant emissions, especially local ones, are important, too (Dutton, 1976). Liao et al., (2020) found that synoptic-scale flow led to an enhanced $PM_{2.5}$ in a coastal area of the Pearl River Delta, while meso/local scale motions led to $PM_{2.5}$ pollution in an inland area. Levy et al. (2010) showed that the concentrations of $NO_x$ and $SO_2$ were higher under the dominance of smaller-scale motions than under larger scale motions.

However, few studies have touched on the impacts of different scales of motion on BC and their effects on radiative efficiency
even though the effects could cause rapid climatic effects due to the patchy and constantly changing distributions (IPCC, 2021).
Topography also plays an important role in air pollution (Zhao et al., 2015). River-valley topography is complicated, and it can
have a considerable influence on air pollution and synoptic patterns of flow (Green et al., 2016; Carvalho et al., 2006). The
pollution levels at cities in river-valleys are not only influenced by general atmospheric dynamics but also strongly impacted
by the local-scale of dynamics (Brulfert et al., 2006). Surface albedo and surface roughness are affected by the complex
topography of river-valley regions, and those physical factors can affect circulation causing changes in pollutant mass
concentrations (Wei et al., 2020). Mountains also significantly affect pollution, and once pollutants are generated or transported
into the river-valley regions, their dispersal can be impeded by the blocking effect of the mountains. Instead of being dispersed,
they can be carried by the airflows over the mountains to converge at the bottom of the valley and increase the pollutants along
the river (Zhao et al., 2015). In this way, pollutants can accumulate in valleys and spread throughout the area, thereby
aggravating pollution. In addition, temperature inversions commonly form in river-valleys during the winter, and that, too, can
aggravate pollution problems (Glojek et al., 2022 and Bei et al., 2016).
Thus, we focused our study on the impacts of different scales of motion on source-specific equivalent BCs (eBCs), and we
evaluated radiative effects of eBCs over a river-valley city. The primary objectives of this study were: (1) to quantify the
contributions of fossil fuel combustion and biomass burning to eBC concentrations, (2) to investigate the impacts of different
scales of motion on the source-specific eBC, and (3) to estimate the radiative effects and the radiative efficiency of the source-
specific eBC under different atmospheric motion scenarios. The study provides insights into the influence of the specified
atmospheric motions on BC and highlights the effects of those motions on the radiative efficiency and potential climatic effects
of the regionally transported BC.
**2 Methodology**
**2.1 Research site**
Baoji is a typical river-valley city, located at the furthest west of the Guanzhong Plain, at an altitude from 450 to 800 m a.s.l.
(Figure S1), Baoji has a complex topography and often suffering from severe pollution in winter. It is surrounded by mountains
to the south, west and north, with the Weihe River as the central axis extending eastward. The shape can be viewed as a funnel,
with large opening to east. The Qinling peaks and the flat Weihe Plain are the main landforms of Baoji. The main peak of the
Qinling Mountains is 3,767 m a.s.l. and it is the highest mountain in the eastern part of mainland China. This terrain causes
divergent flow at local scales, which can impact pollution levels (Wei et al., 2020). Baoji also is an important railway
intersection in China, connecting six railways to the north-west and southwest China. Pollutant levels can be high and pollutants
are not easy to be dispersed in the city due to its special topographic conditions, dense population (total population of 0.341
million, with 63.5% population living in the downtown aera and population density of 6003 people per km$^2$ in 2019
(http://tjj.shaanxi.gov.cn/upload/2021/zk/indexch.htm and https://data.chinabaogao.com/hgshj/2021/042053X932021.html),
and impacts from major highway and railway networks.
The sampling site was on the rooftop of a building at Baoji University of Arts and Sciences (34°21′16.8″N, 107°12′59.6″E,
569 m a.s.l.) surrounded by commercial and residential buildings, highways, and a river, there were no major industrial emission
sources nearby. The main sources of BC in Baoji were the domestic fuel (coal and biomass) burning as well as the motor
vehicle emissions (Zhou et al., 2018; Xiao et al., 2014). Open fire also can be sources for BC, but there were limited fire found
scattered around the site (Figure S2). The meteorological conditions at Baoji for the four seasons are listed in Table S1, and the
wind roses for the different seasons are shown in Figure S3(data are from the Meteorological Institute of Shaanxi Province).
**2.2 Sampling and laboratory measurements**
eBC and the absorption coefficients ($b_{abs}$) at 370, 470, 520, 590, 660, 880, and 950 nm wavelength were measured using an
AE33 aethalometer (Magee Scientific, Berkeley, CA, USA) equipped with a $PM_{2.5}$ cut-off inlet (SCC 1.829, BGI Inc. USA)
that had a time resolution of 1 min. A Nafion® dryer (MD-700-24S-3; Perma Pure, Inc., Lakewood, NJ, USA) with a flow rate
of 5 L $min^{-1}$ was used to dry the $PM_{2.5}$ before the measurement. Briefly, the particles were dried by the Nafion® dryer before
being measured with the AE33 aethalometer, and the deposited particles were irradiated by light-emitting diodes at seven
wavelengths ($\lambda$ = 370, 470, 520, 590,660, 880, and 950 nm), and the light attenuation was detected. The non-linear loading
issue for filter-based absorption measurement was accounted for in the AE33 by a technique called dual-spot compensation.
The quartz filter (PN8060) matrix scattering effect was corrected by using a factor of 1.39. More details of AE33 measurement
techniques can be found in Drinovec et al. (2015).
The scattering coefficient ($b_{scat}$) at a single (525) nm wavelength was measured with the use of a nephelometer (Aurora-1000,
Ecotech, USA) that had a time resolution of 5 min. The nephelometer and aethalometer operated simultaneously and used the
same $PM_{2.5}$ cyclone and Nafion® dryer. The calibration was conducted based on the user guide with a calibration gas R-134.
Zero calibrations were conducted every other day by using clean air without particles. The ambient air was drawn in through a
heated inlet with a flow rate of 5 L $min^{-1}$. The relative humidity remained lower than 60%.
$PM_{2.5}$ samples were collected for every 24 hours (h) from 10 a.m. local time to the 10 a.m. the next day from 16[th] November
2018 to 21[st] December 2018 with two sets of mini-volume samplers (Airmetrics, USA), one using quartz fiber filters (QM/A;
Whatman, Middlesex, UK) and the other with Teflon® filters (Pall Corporation, USA), both with a flow rate of 5 L $min^{-1}$. The
samples were kept in a refrigerator at 4°C before analysis. The mass concentration of $K^+$ in the $PM_{2.5}$ quartz sample was
extracted in a separate 15 mL vials containing 10 mL distilled deionized water (18.2 MΩ resistivity). The vials were placed in
an ultrasonic water bath and shaken with a mechanical shaker for 1 h to extract the ions, which and determined by a Metrohm
940 Professional IC Vario (Metrohm AG., Herisau, Switzerland) with Metrosep C6-150/4.0 column (1.7 mmol/L nitric
acid+1.7 mmol/L dipicolinic acid as the eluent) for cation analysis. A group of elements (i.e. Mg, Al, Si, S, Cl, Ca, V, Mn, Fe,
Ni, Cu, As, Se, Br, Sr, Pb, Ga, and Zn) on the Teflon® filters was were determined by energy-dispersive x-ray fluorescence
(ED-XRF) spectrometry (Epsilon 4 ED-XRF, PANalytical B.V., Netherlands). The X-rays were generated from a gadolinium
anode on a side-window X-ray tube. A spectrum of the ratio of X-ray and photon energy was obtained after 24 minutes of
analysis for each sample with each energy peak characteristic of a specific element, and the peak areas were proportional to the
concentrations of the elements. Quality control was conducted on a daily basis with a test standard sample.
Organic carbon (OC) and elemental carbon (EC) in each sample were determined with the use of a DRI Model 2001
Thermal/Optical Carbon Analyzer (Atmoslytic Inc., Calabasas, CA, USA). The thermal/optical reflectance (TOR) method and
IMPROVE_A protocol were used for analysis. A punch of a quartz filter sample was heated at specific temperatures to obtain
data for four OC fractions and three EC fractions. Total OC was calculated by summing all OC fractions and the pyrolytic
carbon (PC). Total EC was calculated by summing all EC fractions minus the PC. Detailed methods and quality
assurance/quality control processes were described in Cao et al., (2003). Primary organic carbon (POC) was estimated by using
the minimum R-squared (MRS) method, which is based on using eBC as a tracer (Text S1). The method uses the minimum $R^2$
between OC and eBC to indicate the ratio for which secondary OC and eBC are independent. A detailed description of the
MRS method can be found in Wu et al., (2016).
Concentration of $NO_x$, wind speed, and direction at 12 ground monitoring sites were downloaded from
http://sthjt.shaanxi.gov.cn/hx_html/zdjkqy/index.html. The wind data at 100 meters (m) above the ground and the planetary
boundary layer height were downloaded from https://rda.ucar.edu/datasets/ds633.0. The data used for the Hybrid Single-
Particle Lagrangian Integrated Trajectory (HYSPLIT) model was downloaded from Global Data Assimilation System
and it had a resolution of 1°×1° (GDAS, https://www.ready.noaa.gov/gdas1.php). The data and main parameters used in
trajectory model are listed in Table S2.
**2.3 Optical source apportionment**
The positive matrix factorization (PMF) model that was used for the optical source apportionment in this study. PMF solves
chemical mass balance by decomposing the observational data into different source profiles and contribution matrices as
follows:
$$X_{ij} = \sum_{k=1}^{p} g_{ik}f_{kj} + e_{ij} \tag{1}$$
where $X_{ij}$ denotes the input data matrix; $p$ is the number of sources selected in the model; $g_{ik}$ denotes the contribution of the
$k^{th}$ factor to the $i^{th}$ input data; $f_{kj}$ represents the $k^{th}$ factor's profile of the $j^{th}$ species; and $e_{ij}$ represents the residual. Both $g_{ik}$ and
$f_{kj}$ are non-negative. The uncertainties of each species and $b_{abs}(\lambda)$ were calculated by the equation recommended in EPA
PMF5.0 user guideline(Norris et al, 2014) as follows:
$$Unc = \sqrt{(error\ fraction \times concentration(or\ ligth\ absorption\ coefficient))^2 + (0.5 \times MDL)^2} \tag{2}$$
$$Unc = \frac{5}{6} \times MDL \tag{3}$$
where MDL is the minimum detection limit of the method. When the concentration of a species was higher than the MDL then
equation (2) was used otherwise equation (3) was used. In equation (2), for calculating the uncertainty of a chemical species,
the error fraction was multiplied the concentration of the species. For calculating the uncertainty of optical data, the error
fractions were multiplied by the light absorption coefficients.
Chemical species data (EC, POC, $K^+$, Mg, Al, Si, S, Cl, Ca, V, Mn, Fe, Ni, Cu, As, Se, Br, Sr, Pb, Ga and Zn) and the primary
absorption coefficients (Pabs) data at λ=370nm,470nm,520nm,660nm, and 880nm were used for PMF analysis. The error
fraction of offline measured data was the difference between multiple measurements of the same sample. The error fraction
used for optical data was 10% based on Rajesh and Ramachandran (2018). PMF solves the equation (1) by minimizing the $Q$
value, which is the sum of the normalized residuals' squares, as follows,
$$Q = \sum_{i=1}^{n} \sum_{j=0}^{n} \left[\frac{e_{ij}}{u_{ij}}\right]^2 \qquad\qquad (4)$$
where $u_{ij}$ represents the uncertainties of each $X_{ij}$ and $Q_{true}/Q_{exp}$ was used as the indicators for the factor number determination.

## 2.4 eBC source apportionment

The quantities of eBC generated from biomass burning versus fossil fuel combustion were deconvolved by an aethalometer
model which uses Beer-Lambert's Law to write the absorption coefficients equations, wavelengths and absorption Ångström
exponents (AAEs) for the two different BC emission sources (Sandradewi et. al., 2008). This approach is widely used for
separating BC from two different sources based on optical data (Rajesh et al., 2018; Kant et al., 2019; Panicker et al., 2010).
However, the traditional aethalometer model could be affected by the light absorbing substances at lower wavelengths such as
dust and secondary aerosol. An improvement to the traditional aethalometer model was made, by explicitly considering the
interference of the $b_{abs}$ at a lower wavelength (370nm) caused by dust and secondary OC. Thus, the calculation of the absorption
and source apportionment was based on the following equations (Wang et al., 2020):
$$\frac{b_{abs}(370)_{fossil}}{b_{abs}(880)_{fossil}} = \left(\frac{370}{880}\right)^{-AAE_{fossil}} \qquad\qquad (5)$$
$$\frac{b_{abs}(370)_{biomass}}{b_{abs}(880)_{biomass}} = \left(\frac{370}{880}\right)^{-AAE_{biomass}} \qquad\qquad (6)$$
$$b_{abs}(880) = b_{abs}(880)_{fossil} + b_{abs}(880)_{biomass} \qquad\qquad (7)$$
$$b_{abs}(370) = b_{abs}(370)_{fossil} + b_{abs}(370)_{biomass} + b_{abs}(370)_{secondary} + b_{abs}(370)_{dust} \qquad\qquad (8)$$
$$eBC_{fossil} = \frac{b_{abs}(880)_{fossil}}{MAC_{BC}(880)_{fossil}} \qquad\qquad (9)$$
$$eBC_{biomass} = \frac{b_{abs}(880)_{biomass}}{MAC_{BC}(880)_{biomass}} \qquad\qquad (10)$$
where $AAE_{fossil}$ and $AAE_{biomass}$ are the AAEs for fossil fuel combustion and biomass burning. These were derived from the
optical source apportionment by using PMF as discussed in section 3.1. Further, $b_{abs}(370)$ and $b_{abs}(880)$ are the total $b_{abs}$
measured by the AE33 at the wavelengths of 370 nm and 880 nm respectively; $b_{abs}(370)_{fossil}$ and $b_{abs}(880)_{fossil}$ are the $b_{abs}$ caused
by emissions from fossil fuel combustion at those two wavelengths; $b_{abs}(370)_{biomass}$ and $b_{abs}(880)_{biomass}$ are the $b_{abs}$ caused by
emissions from biomass burning at those two wavelengths; $b_{abs}(370)_{dust}$ refers to the $b_{abs}$ contributed by mineral dust at the
wavelength of 370 nm, which was derived from the result of optical source apportionment; $b_{abs}(370)_{secondary}$ refers to the $b_{abs}$
caused by the secondary aerosols at the wavelength of 370 nm, which was calculated by the minimum $R$-squared approach with
eBC as a tracer (Text S1, Wang et al., 2019); $eBC_{fossil}$ and $eBC_{biomass}$ are the eBCs from fossil fuel combustion and biomass
burning; and $MAC_{BC}(880)_{fossil}$ and $MAC_{BC}(880)_{biomass}$ are the mass absorption cross-sections of $eBC_{fossil}$ and the mass
absorption cross-section of $eBC_{biomass}$ at the wavelength of 880 nm respectively, which were based on the PMF results for the
optical source apportionments.

**2.5 Indicators for the different scales of motion**

The mathematical definitions of airflow condition proposed by Allwine and Whiteman (1994) were used in this study. The definitions quantify the flow features integrally at individual stations. Three variables were quantified, namely the actual wind run distance ($S$) which is the scalar displacement of the wind in 24 h (i.e. the accumulated distance of the wind), the resultant transport distance ($L$) which is the vector displacement of the wind in 24 h (i.e. the straight line from the starting point to the end point), and the recirculation factor ($R$) is based on the ratio of $L$ and $S$ which indicates the frequency of the wind veering in 24 h. The influences of different scales of atmospheric motions were assessed based on the method proposed by Levy et al., (2010), and for this, we used wind data at 100 m above the sampling site and the wind data from 12 monitoring stations at ground level (~15m) to indicate the different scales of motions. The winds at the surface monitoring stations were expected to be more sensitive to local-scale turbulence and convection than the winds at 100 m. With less influence from the surface forces, the indicators at 100 m would be more sensitive to larger scales of motion. The equations used as follows:

$$L_{n\tau/bj} = T\left[\left(\sum_{j=i}^{i-\tau+1} u_i\right)^2 + \left(\sum_{j=i}^{i-\tau+1} v_i\right)^2\right]^{1/2} \tag{11}$$

$$S_{n\tau/bj} = \sum_{j=i}^{i-\tau+1}(u_j{}^2 + v_j{}^2)^{1/2} \tag{12}$$

$$R_{n\tau/bj} = 1 - \frac{L_{i\tau}}{S_{i\tau}} \tag{13}$$

where $T$ is the interval of the data (i.e., 60 min), $i$ is the $i^{th}$ the ending time step data, $\tau$ is the integration time period of the wind run (24 h), i-$\tau$+1 represents the data at the start time, and $n$ is the number of monitoring stations (a total of 12 in this study). The quantities $u$ and $v$ are the wind vectors. Using the wind data from the 12 monitoring stations covering Baoji, the $L$ and $S$ values at the 12 different sites at ground level were calculated. $L_{n\tau}$ and $S_{n\tau}$ represent the resultant transport distance and the actual wind run distance at the $n^{th}$ (n = 1 to 12) monitoring station at ground level; $R_{n\tau}$ is the recirculation factor at the $n^{th}$ monitoring station which is calculated based on $L_{n\tau}$ and $S_{n\tau}$; $L_{bj}$, and $S_{bj}$ are the resultant transport distance and the actual wind run distance at 100 m height above the ground. These represent the flow characteristics in higher atmosphere at the study site, and they were calculated by using the wind data at 100 m height. The recirculation factor ($R_{bj}$) was calculated for a height of 100 m.

As explained in Levy et al., (2010), if local-scale motions are strong and regional-scale motions are weak, the variations in winds at each station would not be likely to be uniform due to differences in local factors, and that would result in relatively large standard deviations ($R_{std}$) for $R_{n\tau}$. By contrast, if the local-scale motions are weak and the regional-scale motions are strong, the wind direction would be likely to be more uniform over a large area, and the $R_{bj}$ and the $R_{std}$ should be relatively smaller.

**2.6 Self-organizing map**

A self-organizing map (SOM) developed by Kohonen (1990) is a type of artificial neural network that is widely used for categorizing high-dimensional data into a few major features (Stauffer et al., 2016 and Pearce et al., 2014). In particular, this approach is widely used for categorizing different meteorological patterns (Liao et al., 2020; Han et al., 2020; Jiang et al., 2017). Unlike traditional dimension reduction methods (e.g., principal component analysis), SOM projects high-dimensional

input data by non-linear projection into user-designed lower-dimensions, which are typically two-dimensional arrays of nodes
(Hewitson and Crane, 2006). The performance of SOM in classifying climatological data has been shown to be robust (Reusch
et al., 2005). Competitive learning algorithms are used to train SOM, and the architecture of SOM consists of two layers; one
is called the input layer and it contains the high dimensional input data. The other layer is the output layer in which the node
number is the output cluster number. The working principle of SOM is to convert high dimensional data with complex
correlations into lower dimensions via geometrical relationships (Ramachandran et al., 2019). After the initial random weights
are generated, the input data are compared with each weight, and the best match is defined as winning. The winning node and
the neighboring nodes close to the winning node will learn from the same inputs and the associated weights are updated. After
multiple iterations, the network to settles into stable zones of features and the weights. More detailed working principles of
SOM can be found Kangas and Kohonen, (1996) and Kohonen et al., (1996).
Comparison between the input data and each weight is made by applying Euclidean distances, the best match is defined by the
following equation:
$\|x - m_c\| = min\{\|x - m_i\|\}$         (14)
where $x$ is the input data, $m_c$ is the best matched weight, $m_i$ is the weights connected with the $i^{th}$ node.
The weights are updated by following equation:
$m_i(t + 1) = m_i(t) + h_{ci}(t)[x(t) - m_i(t)]$         (15)
where the $m_i(t + 1)$ is the $i^{th}$ weight at t+1 time, $m_i(t)$ is the $i^{th}$ weight at t time, the $h_{ci}(t)$ is the neighborhood kernel defined
over the lattice points at t time, and $c$ is the winning node location.
SOM was used to categorize the daily atmospheric motions during the study period and to explore the influences of different
scales of motion on source-specific eBC. Hourly averages of three sets of data ($R_{std}$, $L_{bj}$, and $S_{bj}$) were input into SOM.
Determining the size of the output map is crucial for SOM (Chang et al 2020 and Liu et al., 2021). To reduce the subjectivity,
the K-means cluster method was used for the decision-making regarding size. The similarity of each item of the input data
relative to the node was measured using Euclidean distance. The iteration number was set to 2000. For each input data item,
the node closest to it would "win out". The reference vectors of the winning node and their neighborhood nodes were updated
and adjusted towards the data. The "Kohonen" package in *R* language (Wehrens and Kruisselbrink, 2019) was used to develop
the SOM model in this study.
**2.7 Estimations of direct radiative effects and heating rate**
The Santa Barbara DISORT Atmospheric Radiative Transfer (SBDART) model was used to estimate the direct radiative effects
(DRE) induced by source-specific eBC. The model has been used in many studies to calculate the DRE caused by aerosols and
BC (Pathak et al., 2010; Rajesh et al., 2018; Zhao et al., 2019). SBDART calculated DRE based on several well-tested physical
models. Details regarding the model were presented in Ricchiazzi et al., (1998). The important input data included aerosol
parameters, including aerosol optical depth (AOD), single scattering albedo (SSA), asymmetric factor (AF) and extinction
efficiency, surface albedo, and atmospheric profile.
The aerosol parameters used in this study were derived by the Optical Property of Aerosol and Cloud (OPAC) model (Hess et
al., 1998) based on the number concentrations of aerosol components. As the study was conducted in an urban region, the urban
aerosol profile was used in OPAC, and it included soot (eBC), water-soluble matter (WS), and water-insoluble matter (WIS).
The number concentrations of soot were derived from the mass concentrations of eBC with the default ratio (5.99E-5 $\mu g\ m^{-3}$/
$particle.cm^{-3}$) in OPAC. The number concentrations of WS and WIS were adjusted until the modeled SSA and $b_{abs}$ at 500nm
in OPAC were close ($\pm 5\%$, see Figure S4) to those values calculated with data from the nephelometer and AE33 ($b_{ext}(520) =$
$b_{scat}(525) + b_{abs}(520)$, SSA= $b_{scat}(525)/b_{ext}(520)$). The DRE of source-specific eBC at the top of atmosphere (TOA) and surface
atmosphere (SUF) were calculated from the difference between the DREs with or without the number concentrations of the
source-specific eBC under clear-sky conditions.
$DRE_{eBC} = (F\downarrow - F\uparrow)_{with\ eBC} - (F\downarrow - F\uparrow)_{without\ eBC}$                 (16)
$DRE_{eBC,ATM} = DRE_{eBC,TOA} - DRE_{eBC,SUF}$                 (17)
where $DRE_{eBC}$ is the DRE of source-specific eBC, $F\downarrow$ and $F\uparrow$ are the downward and upward flux, $DRE_{eBC,ATM}$ is the DRE of
the source-specific eBC for the atmospheric column, that is, the DRE at the top of the atmosphere ($DRE_{eBC,TOA}$) minus that at
the surface ($DRE_{eBC,SUF}$).
**3 Results and discussion**
**3.1 Calculation of eBC$_{fossil}$ and eBC$_{biomass}$**
The PMF model was used for the optical source apportionment, and those results were used to obtain the site-specific AAEs
and MACs, which in turn were used to calculate the source-specific eBC with the improved aethalometer model. For every
solution, PMF was run 20 times. The $Q_{true}/Q_{exp}$ ratios from the 2- to 7-factor solutions were examined, and the values of a 4-
factor solution were found most stable compared with others because the $Q_{true}/Q_{exp}$ values did not drop appreciably after the
addition of one more factor (Figure S5). Based on these results, the 4-factors solution was determined to be the most
interpretable. Two diagnostic methods, Bootstrap (BS) and Displacement (DISP) (Norris et al, 2014; Brown et al. 2015) were
used to validate the robustness and stability of the results. The BS method was used to assess the random errors and partially
assess the effects of rotational ambiguity while DISP was used to evaluate rotational ambiguity errors. The results of the BS
and DISP analyses showed that there was no swap for the 4-factor solution (Table S3). The modelled primary $b_{abs}(\lambda)$ were well
correlated (r = 0.95–0.96, slope = 0.90~0.95, p < 0.01, Figure S6) with their observed counterparts, which suggested that the
modelling performance of PMF5.0 was good. The factor profiles obtained from the PMF are shown in Figure 1.
The first factor (PC1) had was featured with high loadings of EC (52%), POC (49%), and V (49%) and moderate loadings of
Mn (33%), Ni (40%), Cu (37%), and Zn (44%). This factor source contributed 27% to 44% of the primary $b_{abs}(\lambda)$. Of the species
with high loadings on PC1, EC has been found to be associated with vehicular emissions due to incomplete fuel combustion
(Cao et al., 2013). V and Ni are commonly detected in the particles emitted by diesel-powered vehicles (Lin et al., 2015 and
Zhao et al., 2021). Mn compounds are commonly used as an antiknock additive for unleaded gasoline to raise octane numbers
and protect the engine (Lewis et al., 2003; Geivanidis et al., 2003); and Cu and Zn are emitted by the combustion of lubricating
oils and from the wear of motor vehicle parts (i.e., brakes and tires) (Thorpe and Harrison, 2008; Song et al., 2006). In addition,
the EC associated with this factor was found well correlated (r = 0.83, p < 0.01, Figure S7) with the daily averaged $NO_x$ which
is a commonly used tracer of vehicular emissions in the urban areas (Zotter et al., 2017). Recent research on the source
contributions of BC emissions has shown that most of BC associated with transportation was emitted by on-road diesel vehicles
in China (Xu et al., 2021). From these results, PC1 was identified as diesel vehicular emissions. The MAC of this factor (MAC
$(880)_{diesel}$) was 6.7 $m^2$ $g^{-1}$. The estimated AAE of this factor ($AAE_{diesel}$) was 1.07 (Figure S8), which is comparable with the
AAE values of vehicle emissions (0.8~1.1) reported in previous studies (Zotter et al., 2017; Kirchstetter et al., 2004).
The second factor (PC2) was characterized by the high loadings of $K^+$ (51%), Cl (79%), and Br (52%) and moderate amounts
of EC (26%), POC (28%), and Pb (30%). Of these, $K^+$ is a widely recognized tracers for the biomass burning emissions (Urban
et al., 2012; Zhang et al., 2015), and high loadings of Cl also can be taken as a signal of biomass burning (Yao et al., 2002;
Manousakas et al., 2017). Previous studies showed that a large quantity of Br was found in biomass burning aerosols was
caused by emissions of $CH_3Br$ emission during combustion (Manö and Andreae, 1994; Artaxo et al.,1998). Particulate matter
emitted from biomass burning typically has substantial amounts of OC and EC (Song et al, 2006), and Pb also has been observed
in biomass-burning aerosols (Amato et al., 2016). Thus, PC2 was identified as emissions from biomass burning. The
contribution of this factor to primary $b_{abs}$(370) was as high as 50%, but only 33% to primary $b_{abs}$(880), and that was likely
caused by the brown carbon which is a typically found in biomass-burning aerosols (Washenfelder et al., 2015; Yan et al.,
2015). The MAC of this factor (MAC $(880)_{biomass}$) was 9.5 $m^2$ $g^{-1}$. The AAE of this factor ($AAE_{biomass}$) was 2.13 (Figure S8),
which is consistent with the wide range of AAEs reported for biomass-burning (1.2~3.5) (Sandradewi et al., 2008; Helin et al.,
2018; Zotter et al., 2017).
The third factor (PC3) had significant loadings of S (64%), Se (98%), As (51%), and Pb (53%) and moderate loadings of Ga
(42%)—all of these elements are commonly associated with coal combustion (Hsu et al., 2016; Tan et al., 2017). For instance,
coal combustion has gradually become the main source of Pb in $PM_{2.5}$ after China began to phase out Pb-containing gasoline
(Xu et al. 2012). Thus, PC3 was assigned to coal combustion. The MAC of this factor (MAC $(880)_{coal}$) was 7.5 $m^2$ $g^{-1}$. This
factor contributed 17%–19% primary $b_{abs}(\lambda)$, and its derived $AAE_{coal}$ was 1.74 (Figure S8) which is close to the AAE found for
coal-chunks (Sun et al., 2017).
The last factor (PC4) was most heavily loaded with Al (68%), Si (76%), Ca (65%), Fe (51%), and Sr (71%). These elements
are typical crustal elements, and they are abundant in mineral dust (Tao et al., 2016; Tao et al., 2017). Minor amounts of EC in
crustal dust could be from other EC that had deposited on the ground and later resuspended together with the dust by natural
or artificial disturbances (e.g., wind and traffic flow). This factor only contributed ~4% of the primary $b_{abs}(\lambda)$. The estimated
$AAE_{dust}$ was 1.78 (Figure S8) which is close to the AAE of mineral dust reported in previous studies ($AAE_{370~950}$ = 1.82, Yang
et al., 2009).
As elaborated above, the $PM_{2.5}$ EC over Baoji was mainly from diesel vehicular emissions, biomass burning, and coal
combustion. The emissions can be further grouped into those from biomass burning and fossil fuel combustion (the sum of
diesel vehicular emissions and coal combustion). Thus, the $AAE_{fossil}$ (1.26) and MAC $(880)_{fossil}$ (7.1 $m^2$ $g^{-1}$) were calculated

was the mass-weighted averages (relative to the total EC) of $AAE_{coal}$ (MAC $(880)_{coal}$) and $AAE_{diesel}$ (MAC $(880)_{diesel}$) (Table S4). The hourly mass concentrations of $eBC_{fossil}$ and $eBC_{biomass}$ were then calculated using the 'aethalometer model' (Eqs. 5–10). The results showed that $eBC_{fossil}$ and $eBC_{biomass}$ were only weakly correlated (r = 0.3, Figure S9), indicating a reasonably good separation, and furthermore, their diel variations showed different patterns (Figure 2).

The mean values of $eBC_{fossil}$ and $eBC_{biomass}$ were 2.46 μg m$^{-3}$ and 1.17 μg m$^{-3}$, respectively. The averaged total eBC mass concentration (± standard deviation) was 3.63±2.73μg m$^{-3}$, and the eBC ranged from varying from 0.39 to 12.73 μg m$^{-3}$ during the study period, The averaged mass concentration was comparable to that in Lanzhou, another river valley city in China, that was sampled in the same season (5.1 ± 2.1, Zhao et al.,2019). The lowest value is comparable to other river valley regions such as in Retje in India (Glojek et al., 2022) or in Urumqi River Valley in China (Zhang et al., 2020), however even the highest concentration was much lower than that in other urban regions (Table S5).

The diel variations of $eBC_{fossil}$ (Figure 2a) showed a bimodal pattern with two peaks at 9 a.m. and 7 p.m local time. which are typical peak commuting hours, indicating that there were strong influences from traffic emissions. Due to the reduced traffic flow from 1 a.m. to 5 a.m., $eBC_{fossil}$ decreased slowly. After 5 a.m. passenger vehicles were allowed on the highways in and near Baoji, and $eBC_{fossil}$ started to rise, probably in response to the increased traffic emissions. As the morning commuter traffic increased, $eBC_{fossil}$ reached its first peak at 9 a.m. From then until 11 a.m., $eBC_{fossil}$ declined only slightly because the wind speeds decreased (Figure 2c), which offset the effects of the decreases in traffic. From 11 a.m. to 3 p.m., the increases in the height of the planetary boundary layer (PBLH) (Figure 2d) led to a rapid decrease in $eBC_{fossil}$. Later the PBLH decreased rapidly, resulting in conditions unfavorable for dispersion, and then $eBC_{fossil}$ rose quickly to the second peak at 7 p.m. After passing the evening peak in traffic, the $eBC_{fossil}$ decreased dramatically.

In contrast, the diel variation of $eBC_{biomass}$ (Figure 2b) showed greater influences from meteorological conditions during the daytime, and $eBC_{biomass}$ showed lower concentrations during the day compared with the night. After 6 p.m., increased biomass burning from cooking and residential heating let to the emission of more $eBC_{biomass}$ and the stable PBLH hindered the dispersion of $eBC_{biomass}$; these two factors caused the $eBC_{biomass}$ to reach its peak at 8 p.m. At night, the downslope winds from the mountains converged in the valley at night time (Oke et al., 2002) and turned easterly, where the land altitude is lower than at Baoji (Zhao et al., 2015). This led to t relatively strong winds (Figure 2c) favored dispersion and caused the measured $eBC_{biomass}$ pollutant levels to decrease.

**3.2 The influence of regional and local atmospheric motion on eBC$_{fossil}$ and eBC$_{biomass}$**

The K-means results showed that the four-category solution was appropriate for interpretation as explained above (see also Figure S10). Thus a 2×2 map size was used for the self-organizing map (SOM). The four featured atmospheric motion categories given by SOM (Figure S11) were identified as follows (feature values are in Table 1):

1. Local-scale dominance (LD): This category featured high $R_{bj}$ and $R_{std}$. As described in section 2.5, high $R_{std}$ indicates greater divergence of $R$ at the 12 stations due to the strong influence of local-scale turbulence and convection. $L_{bj}$ and $S_{bj}$ were shorter than 130 km implying stagnation (Allwine and Whiteman, 1994).

2. Local-scale strong and regional-scale weak (LSRW): For this group, $L_{bj}$ and $S_{bj}$ were longer than those for LD, and $R_{std}$
was slightly lower than that in LD.

3. Local-scale weak and regional-scale strong (LWRS): As the values suggest, both $R_{bj}$ and $R_{std}$ were lower than those in LD
and LSRW, especially $R_{bj}$. This suggests the winds veered less frequently and the differences of $R$ found in 12 stations
were smaller than in the two situations above. This situation shows that the influence of the regional-scale motion was
greater than that for the previous two categories.

4. Regional-scale dominance (RD): In this category, wind direction at the study site was nearly uniform (extremely low $R_{bj}$)
suggesting good ventilation (Allwine and Whiteman, 1994). The differences among $R$ found at the 12 stations were even
smaller than for the LWRS group, implying a strong increased influence of regional-scale motions. Indeed, the influence
of regional-scale motions far outweighed the local ones for this category, and therefore, this group was considered to be
dominated by strong regional-scale motions.

As shown in Table 1, the SOM classified 40% of cases were classified as LD, 29% were classified into RD, 17% and 14%
were assigned into LSRW and LWRS respectively. These results indicate that most winter days in Baoji were strongly
influenced by local-scale motions. Under LD, the average mass concentration of eBC$_{fossil}$ ($3.08 \pm 2.07$ μg m$^{-3}$) and eBC$_{biomass}$
($1.52 \pm 1.19$ μg m$^{-3}$) were the highest among all four atmospheric categories noted above and over half (60% for eBC$_{biomass}$ and
55% for eBC$_{fossil}$) of the high values (75$^{th}$ to 100$^{th}$ percentile) were found in this category (Figure 3). In addition, as shown in
Figure 3, the vast majority of the high values are located in the zone indicating air stagnation ($S_{bj} \leq 130$km, shaded yellow).
One difference that the 75$^{th}$ to 100$^{th}$ percentile eBC$_{biomass}$ tended to cluster at $R_{bj} \leq 0.2$ indicates that under LD circumstances,
pollutants were likely coming from the same directions as where the main pollution sources were agglomerated, but eBC$_{fossil}$,
in contrast, evidently originated from more scattered locations ($R_{bj} \geq 0.4$). Under LSRW, the averaged mass concentrations of
eBC$_{fossil}$ and eBC$_{biomass}$ were $2.79 \pm 1.73$ μg m$^{-3}$ and $1.06 \pm 0.83$ μg m$^{-3}$ respectively (Table 1), which were both lower than those
for the LD situation. When the regional scale of motion became stronger (i.e., LWRS and RD), the average mass concentration
of eBC$_{fossil}$ ($2.15 \pm 1.62$ μg m$^{-3}$ and $1.69 \pm 1.36$ μg m$^{-3}$) and eBC$_{biomass}$ ($0.86 \pm 1.58$ μg m$^{-3}$ and $0.93 \pm 0.72$ μg m$^{-3}$) were lower,
presumably because strong winds cause the pollutants to mix with cleaner air. Interestingly, 19% of the total 75$^{th}$ to 100$^{th}$
percentile eBC$_{biomass}$ was found under RD, and 55% of that was when ventilation was good ($S_{bj} \geq 250$km, $R_{bj} \leq 0.2$, Figure 3,
shaded grey). These findings imply that the high mass concentrations of eBC$_{biomass}$ were carried by regional-scale airflow to
the site.
Figure 4 portrays the mass concentrations of eBC$_{fossil}$ and eBC$_{biomass}$ during the daytime and night time respectively under the
four atmospheric motion categories specified earlier. As shown in Figure 4 (a) and (c), the mean values of both types of source-
specific eBCs during daytime were the highest ($3.02 \pm 2.12$ μg m$^{-3}$ and $1.15 \pm 0.8$ μg m$^{-3}$) under LD and the lowest ($1.36 \pm$
$1.00$ μg m$^{-3}$ and $0.58 \pm 0.53$ μg m$^{-3}$) under RD. Meanwhile, the average mass concentrations of both types of eBC decreased
when the influences of the regional scale of atmospheric motion getting were stronger. This suggests that eBC pollution was
apt to accumulated under the dominance of local-scale motions and dispersed under the dominance of regional-scale motions
during the daytime. Similar to the variations in the daytime, the mean values of eBC$_{fossil}$ ($3.00 \pm 2.04$ μg m$^{-3}$) and eBC$_{biomass}$
($1.76 \pm 1.33$ μg m$^{-3}$) under LD were also the highest during the night. However, unlike eBC$_{fossil}$, the mass concentrations of
eBC$_{biomass}$ did not decrease when the influence of regional-scale atmospheric motions was stronger (Figure S12). The mean
value of $eBC_{biomass}$ under RD was the second highest ($1.17 \pm 0.73$ µg m$^{-3}$). The nocturnal PBHL was higher than 100m (Figure
S13) for the RD group, and therefore, the high nocturnal $eBC_{biomass}$ may have been caused by the $eBC_{biomass}$ transported to the
site from upwind regions.

**3.3 Impacts of air mass directions**

Atmospheric motions can not only cause the dispersal of pollution but also bring polluted air to the site from distant sources.
Indeed, air mass movements can mean the difference between no pollution and severe pollution at a receptor site. To examine
the impacts caused by air masses from different directions, the hourly 24h-back trajectories were calculated at 100 m above the
ground using the Hybrid Single-Particle Lagrangian Integrated Trajectory model (Draxler and Hess, 1998, Text S2). Then the
trajectories were clustered by using an angle-based distance statistics method (Text S2) to show the general directional features.
This method determines the direction from which the air masses reach the site and has been widely used for air mass trajectory
clusters. A detailed method description can be found in Sirois and Bottenheim (1995). Three air-mass trajectory clusters were
identified (Figure S14), 45% of total trajectories associated with Cluster No.1, which originated from the north. Cluster No.2
accounted for 36% of the trajectories, and those were from the east direction while Cluster No.3 composed 19% of the total
trajectories and displayed origins from southwest.
Hourly trajectories were assigned into the four featured atmospheric motions. The varying concentrations of the source-specific
eBCs associated with different clusters indicate the divergent impacts of air mass direction on the pollution level at the sampling
site. As shown in Table 1, LD was mainly connected with the air masses from Cluster No.2 (52%) and Cluster No.1 (45%).
The average mass concentrations of $eBC_{fossil}$ and $eBC_{biomass}$ associated with Cluster No.1 were $2.82 \pm 1.59$ µg m$^{-3}$ and $1.34 \pm$
$1.07$ µg m$^{-3}$. In comparison, Cluster No.2 was associated with a higher mean $eBC_{fossil}$ ($3.2 \pm 1.73$ µg m$^{-3}$) and the highest mean
$eBC_{biomass}$ ($1.72 \pm 1.29$ µg m$^{-3}$) of the three clusters. This could be attributed to more intensive emissions in the eastern parts of
Baoji because 75% of the total population of Baoji is located in this area
(http://tjj.baoji.gov.cn/art/2020/10/15/art_9233_1216737.html, accessed on 25 September 2021, in Chinese). Several highways
and railways are located in the south and southwest of Baoji, but the population is sparse with only ~4% of the total population
residing in those areas. Thus, Cluster No.3 was associated with the highest mean $eBC_{fossil}$ concentration ($3.64 \pm 0.67$ µg m$^{-3}$)
but the lowest mean $eBC_{biomass}$ ($0.67 \pm 0.87$ µg m$^{-3}$). It is important to point out, however, that only 3% of the total trajectories
came from this cluster.
Under LSRW, 56% of the trajectories were from Cluster No.1, 33% from Cluster No.2, and 11% from Cluster No.3. Although
the total averaged mass concentrations (Table 1) of two types of eBC generally showed that the regional-scale motions favored
dissipation of eBC compared with LD, the $eBC_{fossil}$ ($3.43 \pm 1.17$ µg m$^{-3}$) associated with Cluster No.2 and $eBC_{biomass}$ associated
with Cluster No.3. ($1 \pm 0.64$ µg m$^{-3}$) were higher by 0.23 µg m$^{-3}$ and 0.33µg m$^{-3}$ respectively relative to the LD case. The rise
of $eBC_{fossil}$ associated with Cluster No.2 was possibly caused by the enhanced regional influence of pollutants brought from
adjacent regions. According to previous studies (Wang et al., 2016; Xu et al., 2016), severe BC pollution in winter is caused
by fossil fuel combustion in Xi'an which is to the east of Baoji. Studies also have reported that high EC emitted from biomass
burning was found to have originated from Sichuan Province (Wu et al., 2020; Cai et al., 2018; Huang et al., 2020) which is to
the southwest of Baoji. Combined with the phenomenon that the mass concentration of $eBC_{biomass}$ associated with Cluster No.3
rose with regional scales of motion, it is reasonable to conclude that the increase of eBC$_{biomass}$ associated with Cluster No.3 was
likely influenced by pollution transport from the southwest.
Under LWRS, 42% of the trajectories were from Cluster No.1., 36% from Cluster No.3, and 22% from Cluster No.2. With
stronger regional scales of motion, the mean values of eBC$_{fossil}$ and eBC$_{biomass}$ associated with all clusters were lower than those
under LD, except for eBC$_{biomass}$ associated with Cluster 3 which increased by 0.52 μg m$^{-3}$. As mentioned before, this increase
could have been caused by regional transport.
In the last category (RD), 41% of the trajectories were from Cluster No.1., 39% from Cluster No.3, and 20% from Cluster No.2.
Similar to the results for LWRS, the average mass concentration of eBC$_{fossil}$ and eBC$_{biomass}$ associated with Cluster No.1 were
only 35% and 48% of the respective values for LD. The average mass concentrations of eBC$_{fossil}$ and eBC$_{biomass}$ associated with
Cluster No.2 were 32% and 51% of the eBC$_{fossil}$ and eBC$_{biomass}$ under LD. As for Cluster No.3, the average mass concentration
of eBC$_{fossil}$ associated with this cluster was also the lowest of all clusters. However, interestingly, the mean value of eBC$_{biomass}$
associated with Cluster No.3 was highest compared with other categories of Cluster No.3. Under strong influences of a regional
scale of motions, the value of eBC$_{biomass}$ was 1.9 times as high as that under LD.
**3.4 Radiative effects**
Figure 5a shows the DREs at top of the atmosphere (DRE$_{eBC, TOA}$), surface (DRE$_{eBC, SUF}$), and the whole atmosphere (DRE$_{eBC,}$
$_{ATM}$) of eBC$_{fossil}$ and eBC$_{biomass}$. The DRE$_{eBC, TOA}$ and DRE$_{eBC, SUF}$ of eBC were 13 W m$^{-2}$ and -22.9 W m$^{-2}$, which were lower
than that reported in Lanzhou (21.8 W m$^{-2}$ and -47.5 W m$^{-2}$ for DRE$_{eBC, TOA}$ and DRE$_{eBC, SUF}$) which is another river valley city
in China (Zhao et al., 2019). This could be due to fact that the eBC mass concentration in Baoji was lower than in Lanzhou
(Table S5). As for the DRE$_{eBC, TOA}$ and DRE$_{eBC, SUF}$ per an unit mass of BC, the results of the two studies were comparable. The
DRE$_{eBC, TOA}$ of eBC$_{fossil}$ (DRE$_{eBCfossil, TOA}$) and eBC$_{biomass}$ (DRE$_{eBCbiomass, TOA}$) were 9.4 ± 7.5 W m$^{-2}$ and 3.6 ± 3.4 W m$^{-2}$ indicating
a warming effect at the top of the atmosphere. The DRE$_{eBC, SUF}$ of eBC$_{fossil}$ (DRE$_{eBCfossil, SUF}$) and eBC$_{biomass}$(DRE$_{eBCbiomass, SUF}$)
were -16.5 ± 13.5 W m$^{-2}$ and -6.4 ± 6.2 W m$^{-2}$ showing a cooling effect at the surface. The DRE$_{eBC, ATM}$ of eBC$_{fossil}$ (DRE$_{eBCfossil,}$
$_{ATM}$) and eBC$_{biomass}$ (DRE$_{eBCbiomass, ATM}$) were 25.9 ± 20.8 W m$^{-2}$ and 10 ± 9.5 W m$^{-2}$ in the atmosphere, indicating a heating
effect.
Figure 5 also shows the DRE$_{eBC, ATM}$ of the source-specific eBC for different atmospheric motions. In general, the changes of
DRE$_{eBC, ATM}$ are in accordance with those of the eBC mass concentrations. The DRE$_{eBCfossil, ATM}$ under LD was the largest with
a mean value of 30.4 ± 23 W m$^{-2}$, followed by LSRW (28.7 ± 20.7 W m$^{-2}$). As the mass concentration of eBC$_{fossil}$ was low
when regional scales of motion were stronger, the DRE$_{eBC, ATM}$ under LWRS and RD were also lower compared with those
under LD or LSRW. By contrast, the DRE$_{eBC, ATM}$ of eBC$_{biomass}$ under LSRW was the highest (11.5 ± 11.8 W m$^{-2}$), but it is only
0.3 W m$^{-2}$ higher than that under LD. When the regional scale of motions became stronger, the DRE$_{eBCbiomass, ATM}$ declined as
expected due to the lower eBC$_{biomass}$ mass concentrations (Figure 4c). The DRE$_{eBC, ATM}$ of eBC$_{biomass}$ under LWRS and RD were
8.6 ± 8.5 W m$^{-2}$ and 7.9 ± 7.4 W m$^{-2}$ respectively.
Although DRE$_{eBC, ATM}$ declined with increased influences from the regional scale of motion, the DRE$_{eBC, ATM}$ efficiency
(DRE$_{eBC, ATM}$ per mass concentration) was found to increase with greater regional-scale motion. Furthermore, the DRE
efficiencies of both types of eBC under LD and LSRW were comparable, around 10 W m$^{-2}$ (Table 2). In contrast, the efficiencies
varied more when the regional-scale motions were stronger. Under LWRS, the efficiencies of $eBC_{fossil}$ and $eBC_{biomass}$ were 13.5
$\pm$ 6.7 and 14.7 $\pm$ 8.1 (W m$^{-2}$)/($\mu$g m$^{-3}$) respectively. Under RD, the efficiencies were even higher, 15.6 $\pm$ 8.9 (W m$^{-2}$)/($\mu$g m$^{-3}$)
for $eBC_{fossil}$ and 15.5 $\pm$ 8.4 (W m$^{-2}$)/($\mu$g m$^{-3}$) for $eBC_{biomass}$, which are > 1.5 times those recorded under LD. The higher eBC
efficiencies may have been caused by the increases in the BC MAC during the regional transport. Studies have confirmed that
the aging processes in the atmosphere can enhance the light-absorbing ability of BC (Chen et al., 2017; Shen et al., 2014), and
regional transport can provide sufficient time for BC aging (Shiraiwa, et al. 2007; Cho et al., 2021). Therefore, the nonlinear
change between mass concentration and DRE efficiency was very likely caused by the strong regional-scale motions that
dispersed fresh BC from local emissions but also brought aged BC to the area from the upwind regions. As a result, under these
conditions, the transported BC reached a receptor site with a higher light-absorbing ability which led to a higher DRE efficiency
of BC at the sampling site. This strongly implies regionally transported BC can greatly perturb climate, particularly at the river-
valley city in our study where dispersion was weak (Zhao et al., 2015; Wang et al., 2013).

## 485    4 Conclusions

This study derived site-specific AAEs using a PMF model for which chemical and optical data collected from a river-valley
city during winter were used as the inputs. Based on the calculated AAEs, source-specific eBCs (i.e., $eBC_{fossil}$ and $eBC_{biomass}$)
were then apportioned using an aethalometer model. Finally, the impacts of different scales of atmospheric motions on the
mass concentrations of the source-specific eBCs and the induced DREs were investigated. Four sources of eBC were identified:
which are diesel vehicular emissions, biomass burning, coal combustion, and mineral dust. The derived AAEs were 1.07 for
diesel vehicular emissions, 2.13 for biomass burning, 1.74 for coal combustion, and 1.78 for mineral dust. The mean values of
$eBC_{fossil}$ and $eBC_{biomass}$ were 2.46 $\mu$g m$^{-3}$and 1.17 $\mu$g m$^{-3}$, respectively.
The self-organizing map indicated that there were four types of atmospheric motions during the sampling period that affected
the mass concentrations of source-specific eBCs. Of these, the local-scale motions were the main influence on most winter
days. The $eBC_{fossil}$ and $eBC_{biomass}$ under those identified atmospheric motions showed that over half of the 75[th] to 100[th] percentile
values for the entire data set were found in LD group (60% for $eBC_{biomass}$ and 55% for $eBC_{fossil}$). This illustrates that the BC
pollution was more severe under the influences of local-scale motion outweighed regional-scale motions. However, even
though regional-scale motions were associated with lower eBCs, 19% of the high values of $eBC_{biomass}$ values occurred under
RD, especially when there was good ventilation. Furthermore, the air masses from different directions also had impacts on the
source-specific eBCs that varied relative to the different atmospheric motions. $eBC_{fossil}$ most likely accumulated under the
influence of strong local-scale motions, but $eBC_{biomass}$ also was found to be increased with the enhanced regional scale of
motions when the air masses from the southwest; this indicates that there were impacts from regional transport.
Similar to the mass concentrations, the DREs of the two types of eBC were both lower when the regional scale of motions were
greater than the local ones. However, the changes in mass concentrations and DREs were not proportionate because the
regional-scale of motions carried the fresh BC away from the local site but brought the aged BCs to the site from the upwind
regions. As a result, the DRE efficiency of eBC was ~1.5 times higher when the regional scale of motion was stronger. This
study showed that different scales of air motions affected the mass concentrations of source-specific eBCs and their DRE
efficiencies. More specifically our study highlights importance of regional transport for the BC radiative forcing and shows

how the enhancement of BC radiative effects caused by aging during regional transport could have especially significant
implications for sites in river valleys. The relationships between BC and atmospheric scales of motion should be evaluated for
other environments besides river valley cities because quantitative information on the relative importance of locally emitted
versus regionally transported materials will be useful for developing pollution controls and for predicting future changes in
climate.

*Data availability*. The data are available from the authors upon request.

*Supplement*. The supplement related to this article is available online.

*Author contributions*. QW and JC designed the study. BZ and SL conducted the field measurements. YQ and JT conducted
data analysis. SL and TZ performed the chemical analysis of filters. HL draft the article and QW revised it. JC and YH
commented on the paper.

*Competing interests*. The authors declare that they have no conflict of interest.

*Acknowledgments*. This research has been supported by the National Natural Science Foundation of China (42192512), the
Key Research and Development Program of Shaanxi Province (2018-ZDXM3-01), the Key Project of CAS (ZDRW-ZS-2017-
6), and the Youth Innovation Promotion Association of the Chinese Academy of Sciences (2019402).

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

**Table 1.** The mass concentration of eBC from fossil fuel combustion ($eBC_{fossil}$) and eBC from biomass burning ($eBC_{biomass}$) associated with different clusters
under four featured atmospheric motions

| Motion category | Local scale dominance (LD) (40%) | | | | Local scale strong and regional scale weak (LSRW) (17%) | | | |
|---|---|---|---|---|---|---|---|---|
| | $L_{bj}$ = 70.9 km, $S_{bj}$ = 107.8 km, $R_{bj}$ = 0.35, $R_{std}$ = 0.25 | | | | $L_{bj}$ = 106.9 km, $S_{bj}$ = 164.8 km, $R_{bj}$=0.33, $R_{std}$ = 0.23 | | | |
| | Cluster 1 | Cluster 2 | Cluster 3 | Total average | Cluster 1 | Cluster 2 | Cluster 3 | Total average |
| Trajectory percentage (%) | 45 | 52 | 3 | 100 | 56 | 33 | 11 | 100 |
| $eBC_{fossil}$ (µg m$^{-3}$) | 2.82[a] ± 1.59[b] | 3.2 ± 1.73 | 3.64 ± 0.67 | 3.08 ± 2.07 | 2.42 ± 1.00 | 3.43 ± 1.17 | 2.89 ± 1.00 | 2.79 ± 1.73 |
| $eBC_{biomass}$ (µg m$^{-3}$) | 1.34 ± 1.07 | 1.72 ± 1.29 | 0.67 ± 0.87 | 1.52 ± 1.19 | 1.0 ± 0.85 | 1.17 ± 0.84 | 1.00 ± 0.64 | 1.06 ± 0.83 |

$L_{bj}$—resultant transport distance, $S_{bj}$—actual wind run distance at 100 m, $R_{bj}$ —recirculation factor at 100 m, $R_{std}$—standard deviation for
recirculation factor. a and b: Mean ± Standard deviation.

Table 1 (continued)

| Motion category | Local scale weak and regional scale strong (LWRS) (14%) | | | | Regional scale dominance (RD) (29%) | | | |
|---|---|---|---|---|---|---|---|---|
| | $L_{bj}$ =159 km, $S_{bj}$ = 183.4 km, $R_{bj}$=0.13, $R_{std}$ = 0.20 | | | | $L_{bj}$ =235.6 km, $S_{bj}$ = 246.4 km, $R_{bj}$= 0.05, $R_{std}$ = 0.18 | | | |
| | Cluster 1 | Cluster 2 | Cluster 3 | Total average | Cluster 1 | Cluster 2 | Cluster 3 | Total average |
| Trajectory percentage (%) | 42 | 22 | 36 | 100 | 41 | 20 | 39 | 100 |
| $eBC_{fossil}$ (µg m$^{-3}$) | 1.32[a] ± 0.67[b] | 2.02 ± 0.73 | 3.16 ± 1.19 | 2.15 ± 1.62 | 1.00 ± 0.64 | 1.02 ± 0.88 | 2.75 ± 1.26 | 1.69 ± 1.36 |
| $eBC_{biomass}$ (µg m$^{-3}$) | 0.67 ± 0.49 | 0.73 ± 0.47 | 1.19 ± 0.60 | 0.86 ± 0.58 | 0.64 ± 0.63 | 0.87 ± 0.69 | 1.26 ± 0.68 | 0.93 ± 0.72 |

$L_{bj}$—resultant transport distance, $S_{bj}$—actual wind run distance at 100 m, $R_{bj}$ —recirculation factor at 100 m, $R_{std}$—standard
deviation for recirculation factor. a and b: Mean ± Standard deviation.
**Table 2.** Direct radiative forcing efficiencies for equivalent black carbon (eBC) from fossil fuel combustion (eBC$_{fossil}$) and the eBC from biomass burning
(eBC$_{biomass}$) under four atmospheric motion categories

| Atmospheric motion category | DRE$_{eBCfossil, ATM}$ efficiency ((W m$^{-2}$)/(µg m$^{-3}$)) | DRE$_{eBCbiomass, ATM}$ efficiency ((W m$^{-2}$)/(µg m$^{-3}$)) |
|---|---|---|
| Local scale dominance (LD) | 10.2[a] ± 4.2[b] | 10.3 ± 4.4 |
| Local scale strong and regional scale weak (LSRW) | 10.6 ±5.7 | 10.2 ± 5.8 |
| Local scale weak and regional scale strong (LWRS) | 13.5 ± 6.7 | 14.7 ± 8.1 |
| Regional scale dominance (RD) | 15.6 ± 8.9 | 15.5 ± 8.4 |

a and b: Mean ± Standard deviation

 **Figure captions:**

**Figure 1.** Four factors identified by source apportionment. Concentration ($\mu g\ m^{-3}$) of the chemical species and primary
absorption coefficients ($p_{abs}$) ($\lambda$) at six wavelengths ($\lambda = 370, 470, 520, 590, 660,$ or $880nm$) ($M\ m^{-1}$) for each source are shown
in grey. The blue square represents the contribution of each chemical species to the four different factors.
**Figure 2.** (a) Diel variations of the eBC from fossil fuel combustion ($eBC_{fossil}$) and (b) the eBC from biomass burning
($eBC_{biomass}$), (c) wind speed ($m\ s^{-1}$) and (d) planetary boundary layer height (m). The black bars of each hourly-averaged point
show the standard deviation.
**Figure 3.** (a) The $75^{th} - 100^{th}$ percentile mass concentrations of the eBC from fossil fuel combustion ($eBC_{fossil}$) and (b) the eBC
from biomass burning ($eBC_{biomass}$) under local scale dominance (LD, red circle), local scale strong and regional scale weak
(LSRW, green circle), local scale weak regional scale strong (LWRS, purple circle) and regional scale dominance (RD, blue
circle). $S_{bj}$ is actual wind run distance at 100m height, $R_{bj}$ is the recirculation factor, the grey area indicates good ventilation
($S_{bj} \geq 250km$, $R_{bj} \leq 0.2$), the yellow area indicates air stagnation ($S_{bj} \leq 130km$).
**Figure 4.** Mass concentrations of the eBC from fossil fuel combustion ($eBC_{fossil}$) and the eBC from biomass burning ($eBC_{biomass}$)
during daytime (a, c) and nighttime (b, d) under local scale dominance (LD); local scale strong and regional scale weak (LSRW);
local scale weak regional strong (LWRS); and regional scale dominance (RD).
**Figure 5.** Direct radiative effect (DRE) of the eBC from fossil fuel combustion ($eBC_{fossil}$) shaded in grey and the eBC from
biomass burning ($eBC_{biomass}$) shaded in yellow (a) in the top atmosphere (TOA), surface (SUF), and the atmosphere atmospheric
column (ATM) and (b) the $DRE_{eBC,ATM}$ of two types of eBC under local scale dominance (LD) shaded in light grey labeled as
LD, local scale strong and regional scale weak (LSRW) shaded in light blue labeled as LSRW, local scale weak regional scale
strong (LWRS) shaded in light grey labeled with LWRS and regional scale dominance (RD) shaded in light blue labeled as RD
(c) DRE efficiencies of $eBC_{biomass}$ (shaded in yellow) and $eBC_{fossil}$ (shaded by grey) in TOA, SUF and ATM (d) DRE efficiencies
of $eBC_{biomass}$ and $eBC_{fossil}$ at ATM under LD (shaded in light grey labeled as LD), LSRW (shaded in light blue labeled as
LSRW), LWRS (shaded in light grey labeled as LWRS) and RD (shaded in light blue labeled with RD).
837 .

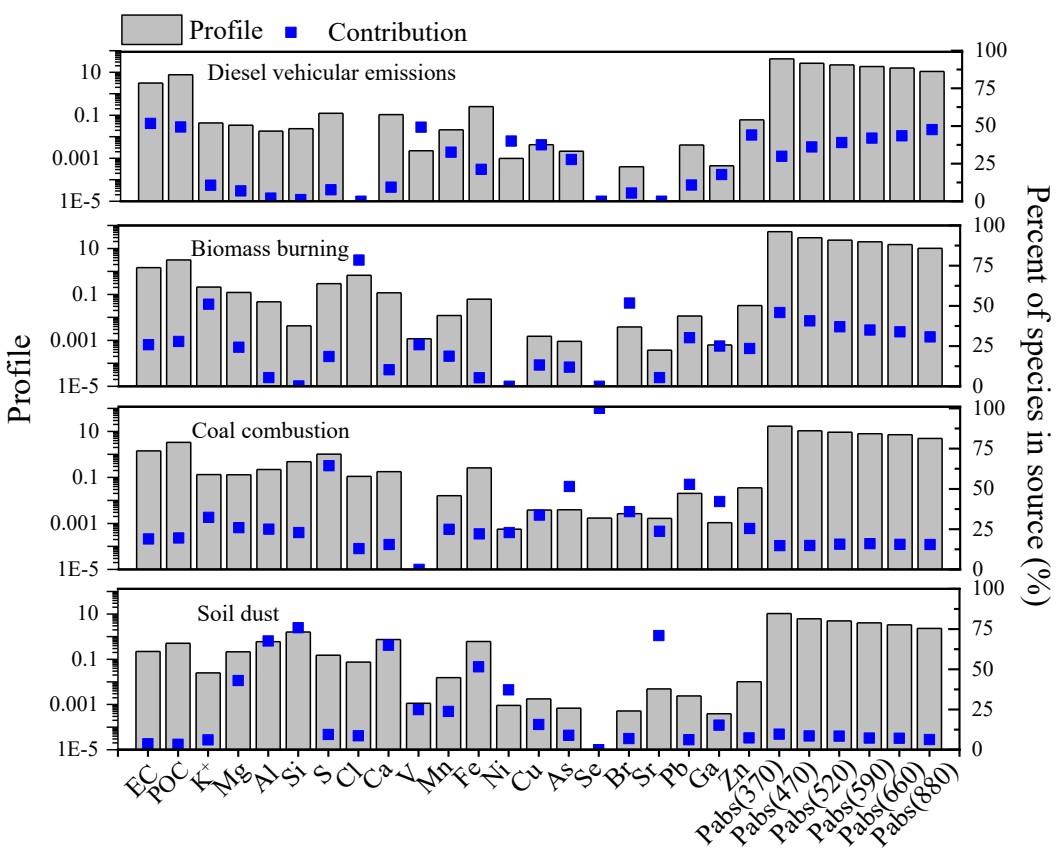


**Figure 1.** Four factors identified by source apportionment. Concentration (μg m$^{-3}$) of the chemical species and primary absorption coefficients (p$_{abs}$) (λ) at six wavelengths (λ = 370, 470, 520, 590, 660, or 880nm) (M m$^{-1}$) for each source are shown in grey. The blue square represents the contribution of each chemical species to the four different factors.

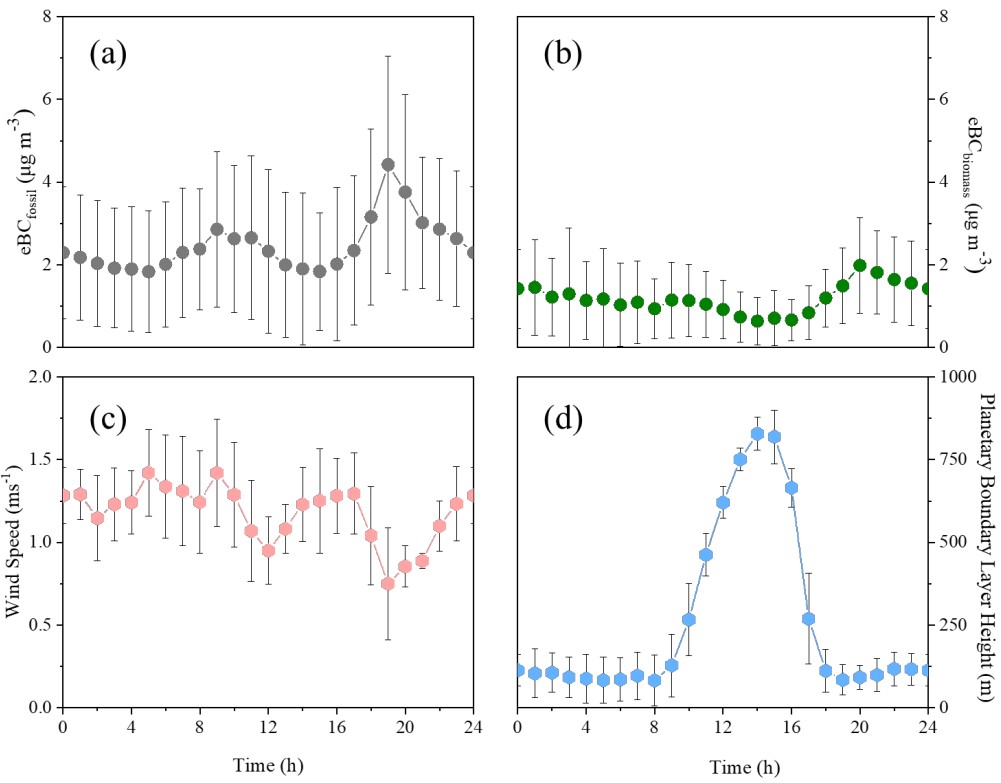


**Figure 2.** (a) Diel variations of the eBC from fossil fuel combustion (eBC$_{fossil}$) and (b) the eBC from biomass burning (eBC$_{biomass}$), (c) wind speed (m s$^{-1}$) and (d) planetary boundary layer height (m). The black bars of each hourly-averaged point show the standard deviation.

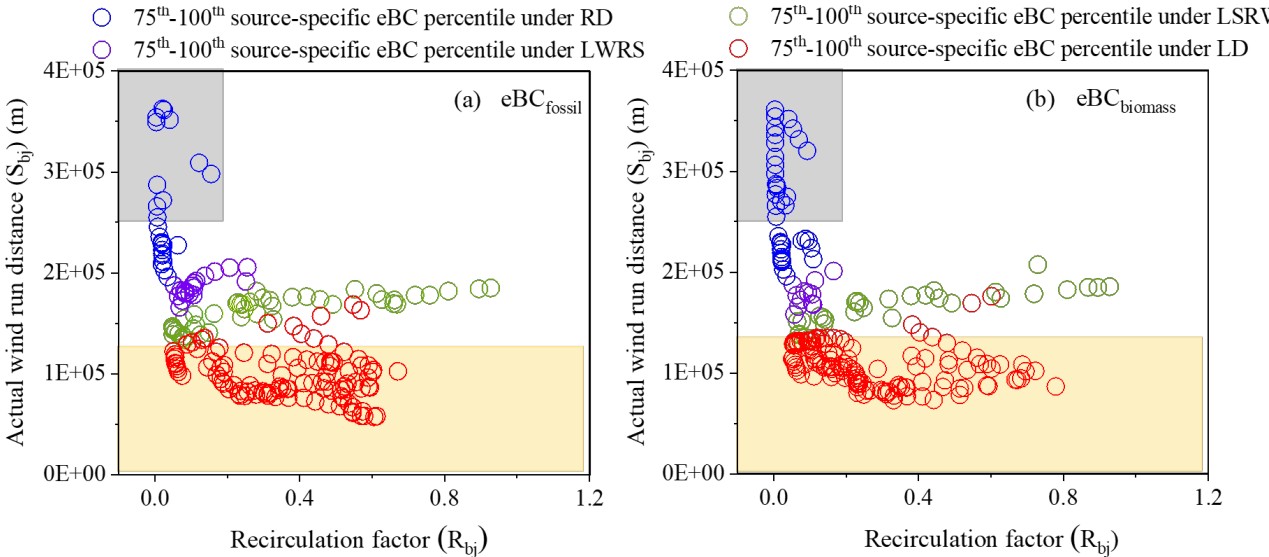


**Figure 3.** (a) The $75^{th}$ – $100^{th}$ percentile mass concentrations of the eBC from fossil fuel combustion (eBC$_{fossil}$) and (b) the eBC
from biomass burning (eBC$_{biomass}$) under local scale dominance (LD, red circle), local scale strong and regional scale weak
(LSRW, green circle), local scale weak regional scale strong (LWRS, purple circle) and regional scale dominance (RD, blue
circle). $S_{bj}$ is actual wind run distance at 100m height, $R_{bj}$ is the recirculation factor, the grey area indicates good ventilation
($S_{bj} \geq 250km$, $R_{bj} \leq 0.2$), the yellow area indicates air stagnation ($S_{bj} \leq 130km$).

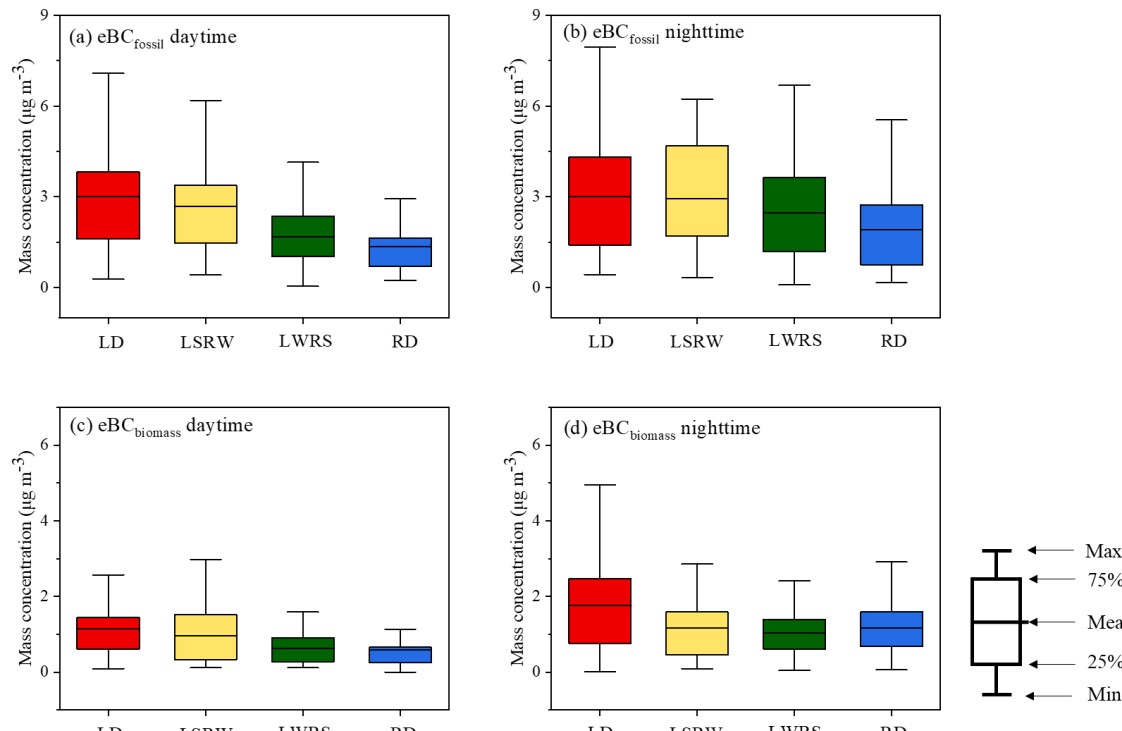


**Figure 4.** Mass concentrations of the eBC from fossil fuel combustion (eBC$_{fossil}$) and the eBC from biomass burning (eBC$_{biomass}$)
during daytime (a, c) and nighttime (b, d) under local scale dominance (LD); local scale strong and regional scale weak (LSRW);
local scale weak regional strong (LWRS); and regional scale dominance (RD).

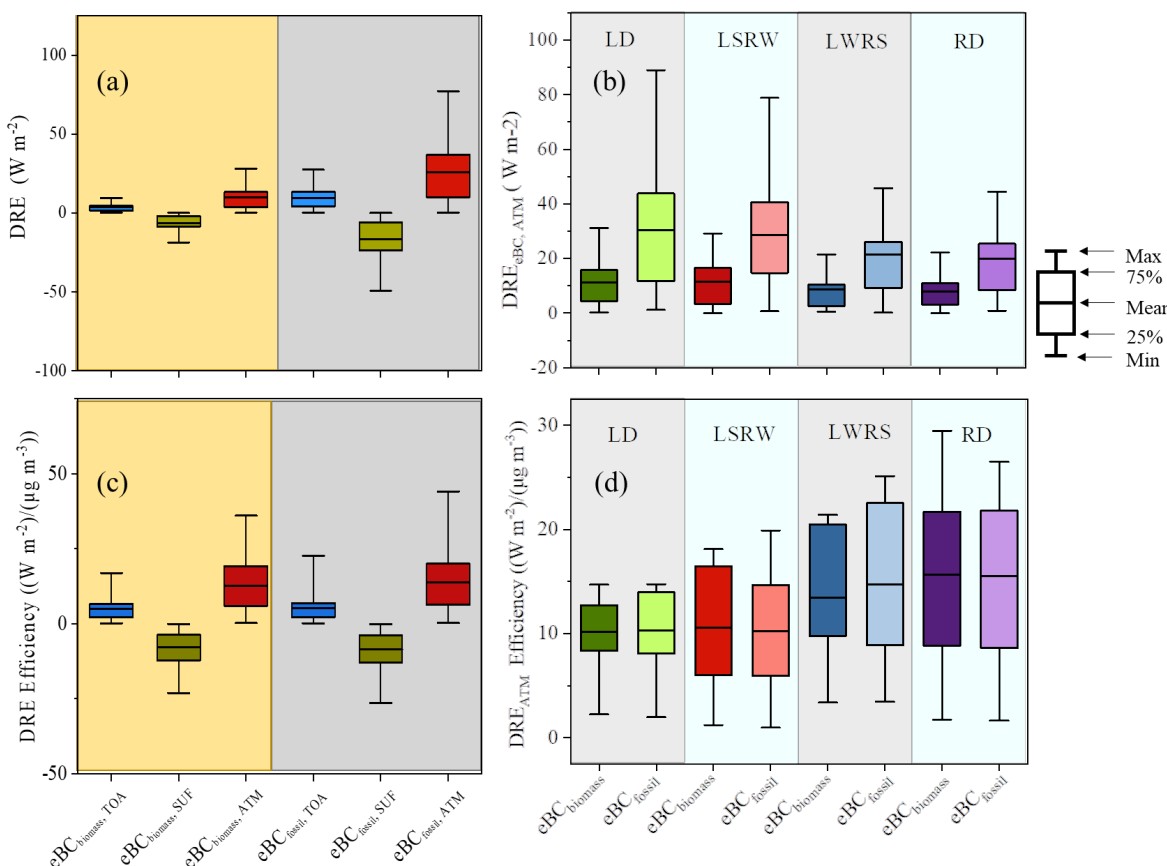


**Figure 5.** Direct radiative effect (DRE) of the eBC from fossil fuel combustion (eBC$_{fossil}$) shaded in grey and the eBC from biomass burning (eBC$_{biomass}$) shaded in yellow (a) in the top atmosphere (TOA), surface (SUF), and the atmosphere atmospheric column (ATM) and (b) the DRE$_{eBC,ATM}$ of two types of eBC under local scale dominance (LD) shaded in light grey labeled as LD, local scale strong and regional scale weak (LSRW) shaded in light blue labeled as LSRW, local scale weak regional scale strong (LWRS) shaded in light grey labeled with LWRS and regional scale dominance (RD) shaded in light blue labeled as RD (c) DRE efficiencies of eBC$_{biomass}$ (shaded in yellow) and eBC$_{fossil}$ (shaded by grey) in TOA, SUF and ATM (d) DRE efficiencies of eBC$_{biomass}$ and eBC$_{fossil}$ at ATM under LD (shaded in light grey labeled as LD), LSRW (shaded in light blue labeled as LSRW), LWRS (shaded in light grey labeled as LWRS) and RD (shaded in light blue labeled with RD).

866