# Peer review of "The impact of atmospheric motions on source-specific black carbon and the induced direct radiative effects over a river-valley region"

_Atmospheric Chemistry and Physics, 2022_

## Author Comment (AC1)

*Comment on acp-2022-26 titled "Review of "The impact of atmospheric motion on source-specific black carbon and the induced direct radiative effect over a river-valley region" by Liu et al.*

**Anonymous Referee #2**

**General comment**

*The site location should be described in more detail, and also earlier on. Some more comments on this issue are below.*

**Response:** We have taken the suggestion to heart and have added more details about the location description. The revised site description now reads:

"Baoji is a typical river-valley city, located at the furthest west of the Guanzhong Plain, at an altitude from 450 to 800 m a.s.l. (Figure S1), Baoji has a complex topography and often suffering from severe pollution in winter. It is surrounded by mountains to the south, west and north, with the Weihe River as the central axis extending eastward. The shape can be viewed as a funnel, with large opening to east. The Qinling peaks and the flat Weihe Plain are the main landforms of Baoji. The main peak of the Qinling Mountains is 3,767 m a.s.l. and it is the highest mountain in the eastern part of mainland China. This terrain causes divergent flow at local scales, which can impact pollution levels (Wei et al., 2020). Baoji also is an important railway intersection in China, connecting six railways to the north-west and southwest China. Pollutant levels can be high and pollutants are not easy to be dispersed in the city due to its special topographic conditions, dense population (total population of 0.341 million, with 63.5% population living in the downtown aera and population density of 6003 people per km$^2$ in 2019 (http://tjj.shaanxi.gov.cn/upload/2021/zk/indexch.htm and https://data.chinabaogao.com/hgshj/2021/042053X932021.html), and impacts from major highway and railway networks.

The sampling site was on the rooftop of a building at Baoji University of Arts and Sciences (34°21′16.8″N, 107°12′59.6″E, 569 m a.s.l.) surrounded by commercial and residential buildings, highways, and a river, there were no major industrial emission sources nearby. The main sources of BC in Baoji were the domestic fuel (coal and biomass) burning as well as the motor vehicle emissions (Zhou et al., 2018; Xiao et al., 2014). Open fire also can be sources for BC, but there were limited fire found scattered around the site (Figure S2). The meteorological conditions at Baoji for the four seasons are listed in Table S1, and the wind roses for the different seasons are shown in Figure S3(data are from the Meteorological Institute of Shaanxi Province)."

**Table R1** The seasonal meteorological data of Baoji

| Season | Temperature (°C) | Relative humidity (%) | Precipitation in last hour (mm) |
|---|---|---|---|
| Winter | 2.7 | 60.6 | 0.025 |
| Spring | 11.5 | 54.9 | 0.042 |
| Summer | 23.7 | 67.1 | 0.139 |
| Autumn | 20.2 | 67.0 | 0.074 |

[Figure]

**Figure R1** Seasonal wind roses for Baoji.

*The English grammar and style could be quite significantly improved. A few examples of some issues are provided in the specific comments below, but many more instances are present in the paper. Maybe some editorial work by a native-speaking scientist could help improve the language and therefore clarity.*

**Response:** Thanks for pointing out those grammatical mistakes in the specific comments and as suggested we had the paper polished by a native English speaker.

*The origin of the biomass burning BC is not well described or discussed. Is this agricultural biomass burning, wildfires, or residential biomass burning? Something else?*

**Response:** The field sampling happened in winter (from 16th November 2018 to 21st December 2018), in Baoji wildfires were comparatively sparser, we screenshotted the fire map (data from MODIS/Aqura, MODIS/Terra and VIIRS 375m/Suomi NPP) from NASA, as shown in Figure R2. Although China has banned agriculture biomass burning due to the air pollution, a lack of strict supervision leads to a few illegal burning in rural areas. Biomass (e.g. straw and crop) is one of the common solid fuel for residential heating and home cooking in winter which causes severe air pollution (Wu et al., 2019). Thus, we believe the biomass burning BC could be from all the three sources but residential biomass burning may take the major responsibility. We have added the potential emission sources of BC at study site in the section 2.1 Research site, see the response above.

[Figure]

**Figure R2.** Map of fire occurrences. The yellow star represents the study site, the red dots represent the fire. The image from © NASA (National Aeronautics and Space Administration) (https://firms.modaps.eosdis.nasa.gov/map).

**Reference:**

Wu S., Zheng, X., You, C., and Wei, C.: Household energy consumption in rural China: Historical development, present pattern and policy implication, Journal of Cleaner Production, 211, 981-991, https://doi.org/10.1016/j.jclepro.2018.11.265, 2019.

*The literature citations are a bit scarce and somewhat biased. A broader representation of the work published in the literature and relevant to the authors' work outside of their own work, would help.*

**Response:** We reviewed more studies and replaced some of the research citation from our work. In addition, we added the comparison of studies on BC pollution and the total DRE of BC in other river valley sites. The revised part is shown was following:

"The mean values of $eBC_{fossil}$ and $eBC_{biomass}$ were 2.46 μg m$^{-3}$ and 1.17 μg m$^{-3}$, respectively. The averaged total eBC mass concentration (± standard deviation) was 3.63±2.73μg m$^{-3}$, and the eBC ranged from varying from 0.39 to 12.73 μg m$^{-3}$ during the study period, The averaged mass concentration was comparable to that in Lanzhou, another river valley city in China, that was sampled in the same season (5.1 ± 2.1, Zhao et al.,2019). The lowest value is comparable to other river valley regions such as in Retje in India (Glojek et al., 2022) or in Urumqi River Valley in China (Zhang et al., 2020), however even the highest concentration was much lower than that in other urban regions (Table S5)."

"The $DRE_{eBC, TOA}$ and $DRE_{eBC, SUF}$ of eBC were 13 W m$^{-2}$ and -22.9 W m$^{-2}$, which were lower than that reported in Lanzhou (21.8 W m$^{-2}$ and -47.5 W m$^{-2}$ for $DRE_{eBC, TOA}$ and $DRE_{eBC, SUF}$) – which is another a river valley city in China (Zhao et al., 2019). This could be due to fact that the eBC mass concentration in Baoji was lower than in Lanzhou (Table S5). As for the $DRE_{eBC, TOA}$ and $DRE_{eBC, SUF}$ per an unit mass of BC, the results of the two studies were comparable."

**Table R2** Mean (range) BC mass concentration in river valley sites worldwide.

| Reference | BC concentration ($\mu g\ m^{-3}$) | Season | Topographic conditions | Altitude | Station type | Year |
|---|---|---|---|---|---|---|
| This study | 3.63±2.73 (0.39~12.73) | November~December (winter) | river valley | 450 to 800 m a.s.l. | urban | 2018 |
| Glojek et al., (2020) | 0.9~40 | December~January (winter) | river valley | 715 m a.s.l. | rural | 2017-2018 |
| Zhao et al. (2015) | 25±11 | January (winter) | river valley | 410 m a.s.l. | urban | 2013 |
| Barman and Gokhale (2019) | 20.58~22.44 | Winter | river valley | | urban | 2016-2017 |
| Zhang et al., 2020 | 0.102~1.525 | Winter | river valley | 2130 m a.s.l. | rural | 2016-2017 |
| Chakrabarty et al., (2012) | 9~41 | January~February (winter) | river valley | | urban | 2011 |
| Zhao et al. (2019) | 5.1 ± 2.1 | December~January (winter) | river valley | | urban | 2018 |
| Tiwari et al., 2016 | 8.19 ±1.39 | December-February (winter) | river valley | 55 m a.s.l. | urban | 2013-2014 |

**Reference:**

Barman, N., and Gokhale, S., Urban black carbon - source apportionment, emissions and long-range transport over the Brahmaputra River Valley, Science of the Total Environment, 693, 133577, https://doi.org/10.1016/j.scitotenv.2019.07.383, 2019

Chakrabarty, R., Garro, M., Wilcox, E., and Moosmuller, H., Strong radiative heating due to wintertime black carbon aerosols in the Brahmaputra River Valley, Geophysical Research Letters, 39, L09804, https://doi.org/10.1029/2012GL051148, 2012.

Glojek, K., Moˇcnik, G., Alas, H., et al., The impact of temperature inversions on black carbon and particle mass concentrations in a mountainous area, Atmos. Chem. Phys., 22, 5577–5601, https://doi.org/10.5194/acp-22-5577-2022, 2022.

Zhang, X., Li, Z., Ming, J., and Wang, F., One-Year Measurements of Equivalent Black Carbon, Optical Properties, and Sources in the Urumqi River Valley, Tien Shan, China, Atmosphere, 11, 478, https://doi.org/10.3390/atmos11050478, 2020.

Zhao, S., Yu, Y., Yin, D., et al., Concentrations, optical and radiative properties of carbonaceous aerosols over urban Lanzhou, a typical valley city: Results from in-situ observations and numerical model, Atmospheric Environment, 213, 470–484, https://doi.org/10.1016/j.atmosenv.2019.06.046, 2019.

Zhao, S., Tie, X., Cao, J & Zhang, Q. (2015), Impacts of mountains on black carbon aerosol under different synoptic meteorology conditions in the Guanzhong region, China. Atmospheric Research, 164-165, 286-296. http://dx.doi.org/10.1016/j.atmosres.2015.05.016

Tiwari, S., Kumar, R., Tunved, P., Singh, S., and Panicker, A., Significant cooling effect on the surface due to soot particles over Brahmaputra River Valley region, India: An impact on regional climate, 2016, Science of the Total Environment, 562, 504–516, http://dx.doi.org/10.1016/j.scitotenv.2016.03.157, 2016.

*Maybe is my lack of familiarity with some of these aspects, but some of the data analysis methods (e.g., SOM, but not only) are not described in sufficient detail. The authors refer to existing literature, but a brief description of the methods' workings, input, outputs, limitations, etc. would help improve the clarity and broaden the audience of the paper.*

**Response:** We are truly grateful for reviewer's comments and suggestion. We revised the methods to provide more details particularly for the optical source apportionment and SOM. Other methods also have been revised according to the reviewer's suggestions. In addition, we have provided information regarding the cluster analysis and minimum R squared method in the supplementary materials. The changes are shown in the revised version. The revised paragraphs now read as follows:

[revised manuscript text omitted]

**"Text S1. Minimum R‑squared method**

The minimum R squared method developed by Wu et al., (2016) was used to separate secondary organic carbon (SOC) from the primary organic carbon (POC). The assumption behind this method is the organic carbon (OC) from non-combustion source is negligible. As explained by Wang et al., (2019), the major non-combustion source is biogenic which is mainly exists in coarse mode. Thus, the non-combustion organic carbon is considered negligible in this study. Therefore, SOC and POC can be separated by using following equations. For each date set, the ratios of OC to eBC and SOC and the $R^2$ between eBC and SOC can be calculated. SOC and eBC are considered independent, so the (OC/eBC)pri should be the value obtained when the $R^2$ between eBC and SOC is minimum.

$$POC = (OC/EC)_{pri} \times EC \tag{S1}$$

$$SOC = OC_{total} - (OC/EC)_{pri} \times EC \tag{S2}$$

where EC in this study is eBC. The (OC/EC)pri is the ratio in freshly emitted OC and EC from combustion sources.

The light absorption at shorter wavelengths (<660nm) is not only from primary light absorbing substances but also from the secondary organic carbon (Wang et al., 2019). The assumption for this method is that the light absorption caused by non-combustion sources is negligible. As mentioned above, most of the biogenic BrC is in coarse mode. Another common light absorbing substance is the $Fe_2O_3$ in the dust, but the impact of that should be limited because the absorption from $Fe_2O_3$ in the dust has been reported to be much smaller than that from BC (Ramachandran and Kedia, 2010). Thus, to separate the secondary light absorption ($b_{abs}(\lambda)_{secondary}$) from the primary light absorption ($b_{abs}(\lambda)_{primary}$), a BC-tracer method coupled with a minimum R-squared method was used. The equations used for the calculation are follows:

$$b_{abs}(\lambda)_{secondary} = b_{abs}(\lambda) - (\frac{b_{abs}(\lambda)}{BC})_{pri} \times BC \tag{S3}$$

$$b_{abs}(\lambda)_{primary} = b_{abs}(\lambda) - b_{abs}(\lambda)_{secondary} \tag{S4}$$

Where $b_{abs}(\lambda)$ is the light absorption at different wavelengths ($\lambda$=370nm, 470nm, 520nm, 590nm, 660nm) measured by AE33, BC is the eBC measured by AE33 at a wavelength of 880nm. The $(\frac{b_{abs}(\lambda)}{BC})_{pri}$ is the ratio of the primary light absorption to the BC mas concentration from combustion sources.

**Text S2. Cluster analysis of air-mass trajectories**

Back trajectories were calculated by using Hybrid Single-Particle Lagrangian Integrated Trajectory (HYSPLIT) model (Draxler and Hess, 1998) developed by the Air Resource Lab (ARL) of the National Oceanic and Atmospheric Administration (NOAA). The model can predict the position of air mass by using mean wind. The back-in-time positions are calculated by reversing the advection equation (Draxler and Hess, 1997). The calculation requires the mean wind, for calculating trajectories, only advection is considered (Stein et al., 2015). The basic equations for trajectory calculation in HYSPLIT are as follows:

$$P^{'}(t + \Delta t) = P(t) + V(P, t) \times \Delta t \qquad (S5)$$

$$P(t + \Delta t) = P(t) + 0.5 \times [V(P, t) + V\left(P^{'}, t + \Delta t\right)] \times \Delta t \qquad (S6)$$

Where $P(t)$ is the initial position, $P^{'}(t + \Delta t)$ is the first guess position, $V$ is the average velocity, $t$ is the time, $\Delta t$ is the time step.

A large number of 24 h trajectories (793) that were retrieved for the study period showed diverse pathways, so in order to find out the representative pathways for those trajectories, a cluster analysis based on an angle-based distance statistics method was conducted. Compared with Euclidean distance, angle-based distance statistics method focuses on the direction of air mass instead of the speed. The angle-based distance statistics method is defined by following equations (Sirois and Bottenheim, 1995):

$$d_{12} = \frac{1}{2}\sum_{i=1}^{n} cos^{-1}(0.5 \times \frac{A_i + B_i - C_i}{\sqrt{A_i B_i}}) \qquad (S7)$$

$$A_i = (X_1(i) - X_0)^2 + (Y_1(i) - Y_0)^2 \qquad (S8)$$

$$B_i = (X_2(i) - X_0)^2 + (Y_2(i) - Y_0)^2 \qquad (S9)$$

$$C_i = (X_2(i) - X_1(i))^2 + (Y_2(i) - Y_1(i))^2 \qquad (S10)$$

Where $d_{12}$ is the average angle between the two backward trajectories, varying between 0 and $\pi$; $X_0$ and $Y_0$ are the position of the receptor site; and $X_1$ ($Y_1$) and $X_2$ ($Y_2$) are the backward trajectories 1 and 2, respectively. In this study, three clusters were chosen as representative of the backward trajectory clusters based on the total spatial variance (TSV) value. The simulation was conducted using the GIS-based TrajStat software (Wang et al., 2009)."

**Specific comments**

**Abstract:**

 *"Black carbon (BC) has a strong light absorption ability and is known as the second strongest light-absorbing substance in the atmosphere after CO2"* *This is debatable.*

**Response:** We have corrected the sentence into "Black carbon (BC) is one of the most important short lived climate forcers, and atmospheric motions play an important role in determining its mass concentrations of pollutants."

In the first paragraph in the introduction part, we also revise the relevant sentence into:

"Black carbon (BC) is produced by the incomplete combustion of biomass and fossil fuels. The BC aerosol has a strong light absorption capacity and can cause heating of the atmosphere. In fact, BC is widely recognized as one of the most important short-lived climate forcers (IPCC, 2021)."

*What does model refer to in "aethalometer model"?*

**Response:** The model refers to the calculation model based on Beer–Lambert's Law. The model contains a few equations relating the absorption coefficients ($b_{abs}$), the wavelengths, and the absorption exponents for conditions of two BC emission sources (Sandradewi et. al., 2008). In this study the two sources were fossil

combustion and biomass burning. We have revised the method section 2.4 to provide a better description and reference. The revised version was provided in response above.

**Reference:**

Sandradewi, J., Prévôt, A. S. H., Weingartner, E., Schmidhauser, R., Gysel, M., and Baltensperger, U.: A study of wood burning and traffic aerosols in an Alpine valley using a multi-wavelength Aethalometer, Atmos. Environ., 42, 101-112, https://doi.org/10.1016/j.atmosenv.2007.09.034, 2008.

*"chemical data and optical data" what kind of data?*

**Response:** The chemical data includes EC, POC, $K^+$, Mg, Al, Si, S, Cl, Ca, V, Mn, Fe, Ni, Cu, As, Se, Br, Sr, Pb, Ga, and Zn. The optical data refers to primary absorption coefficient ($b_{abs}$ ($\lambda$)) at six wavelengths ($\lambda$ = 370, 470, 520, 590, 660, and 880 nm). To make this clear, we have revised the relevant sentence into

"Equivalent BC (eBC) source apportionment was based on an aethalometer model with the site-dependent absorption Ångström exponents (AAEs) and the mass absorption cross-sections (MACs) retrieved using a positive matrix factorization (PMF) model based on observed chemical components (i.e. EC, POC, $K^+$, Mg, Al, Si, S, Cl, Ca, V, Mn, Fe, Ni, Cu, As, Se, Br, Sr, Pb, Ga, and Zn) and primary absorption coefficients at selected wavelengths from $\lambda$ = 370 to 880nm."

*"The derived AAEs" over what wavelength range?*

**Response:** The AAEs were obtained by the power fit of $b_{abs}$ from 370nm, 470nm, 520nm, 660nm and 880nm. The fitting is shown in figure R3. We also added this figure into supplement material and revised the relevant sentence shown below:

"The derived AAEs from 370 to 880nm were 1.07 for diesel vehicular emissions, 2.13 for biomass burning, 1.74 for coal combustion, and 1.78 for mineral dust. The mean values for eBC$_{fossil}$ and eBC$_{biomass}$ were 2.46 μg m$^{-3}$ and 1.17 μg m$^{-3}$ respectively."

[Figure]

**Figure R3** Light absorption ($b_{abs}(\lambda)$) for diesel vehicular emissions, biomass burning, coal combustion, and mineral dust. The dashed lines are the power law fits.

*"four featured atmospheric motions categories" what the four categories are remains a mystery until later, please provide a brief description here because the abstract should be self-contained.*

**Response:** Thanks for the suggestion, we have revised this sentence to add more information on the four categories as follows:

"Wind run distances and the vector displacements of the wind in 24 h were used to construct a self-organizing map, from which four atmospheric motions categories were identified (local-scale dominant, local-scale strong and regional-scale weak, local-scale weak and regional-scale strong and regional-scale dominant)."

*"The trajectory clusters" what trajectories? How were those determined?*

**Response:** The trajectories in this study reconstruct the path the air mass moved through in time and space. The back trajectories were calculated by using Hybrid Single-Particle Lagrangian Integrated Trajectory (HYSPLIT) model (Draxler and Hess, 1998). The model can predict the position of air mass by using the mean winds. The backwards-in-time position is calculated by reversing the advection equation (Draxler and Hess, 1997). The calculation needs the mean wind, for running trajectories, and only advection is considered (Stein et al., 2015). The basic equations for trajectory calculation in HYSPLIT are as followings:

$$P'(t + \Delta t) = P(t) + V(P, t) \times \Delta t \tag{R1}$$

$$P(t + \Delta t) = P(t) + 0.5 \times [V(P, t) + V(P', t + \Delta t)] \times \Delta t \tag{R2}$$

Where $P(t)$ is the initial position, $P'(t + \Delta t)$ is the first guess position, $V$ is the average velocity, $t$ is the time, $\Delta t$ is the time step.

To this clear, we have revised the sentence into "Cluster analysis for the back trajectories of air mass calculated by Hybrid Single-Particle Lagrangian Integrated Trajectory model at the study site indicated that the directions of air flow can have different impacts for different scales of motion." And added the method description above into the supplementary material (Text S2) and provided a Table (Table S2) to show the input data, main parameters used in this study as follow:

**Table R3** Data and parameters used in HYSPLIT model

| Items | Data/parameters |
|---|---|
| Model | HYSPLIT |
| Meteorological data | GDAS data, $1° \times 1°$, 23 vertical levels, 3 hourly |
| Backward period | 24h |
| Footprint level | 100 m above the ground |
| Receptor site location | 34°21′16.8″N, 107°12′59.6″E |

**Reference:**

Stein, A., Draxler, R., Rolph, G., Stunder, B., Cohen, M., and Ngan, F.,: NOAA'S HYSPLIT Atmospheric Transport and Dispersion Modeling System, Bull. Amer. Meteor. Soc., 96, 2059-2077, https://doi.org/10.1175/BAMS-D-14-00110.1, 2015.

Draxler, R., and Hess, G.: An overview of the HYSPLIT_4 modelling system for trajectories, Aust. Meteorol. Mag., 47, 1998.

*Lines 30-31: I don't understand the sentence "This study revealed the disproportional change between BC mass concentration and its DRE."*

**Response:** We apologize for the confusing expression, and we would like to provide a further explanation and have revised the sentence to make it clear.

If the light absorbing ability is independent of the patterns of the motion (which indicates a possible path and distance the BC moved in atmosphere), the value of DRE per unit mass of BC should be the same for all cases. However, in this study, both DREs and the mass concentrations of $eBC_{fossil}$ and $eBC_{biomass}$ changed. If we normalize DRE by dividing the mass concentration of eBC, we can see that a unit of eBC of both types associated with higher DRE under local scale weak and regional scale strong motion (LWRS) and regional scale dominance motion (RD) as shown in Table R4. If we take local scale dominance (LD) as a reference case to calculate the difference between averaged BC mass concentration (or averaged DRE) under LD and other cases (LSTW, LWRS and RD). It is apparent that the BC mass concentrations decreased more than DRE did (Table R5).

To avoid confusion, we revised the sentence into "The finding that the DRE efficiency of BC increased during the regional transport suggested significant consequences in regions downwind of pollution sources and emphasizes the importance of regionally transported BC for potential climatic effects." And "Similar to the mass concentrations, the DREs of the two types of eBC were both lower when the regional scale of motions were greater than the local ones. However, the changes in mass concentrations and DREs were not proportionate because the regional-scale of motions carried the fresh BC away from the local site but brought the aged BCs to the site from the upwind regions. As a result, the DRE efficiency of eBC was ~1.5 times higher when the regional scale of motion was stronger."

**Table R4.** Direct radiative forcing efficiencies for equivalent black carbon (eBC) from fossil fuel combustion (eBCfossil) and the eBC from biomass burning

| Atmospheric motion category | $DRE_{eBCfossil, ATM}$ efficiency[a] $((W\ m^{-2})/(\mu g\ m^{-3}))$ | $DRE_{eBCbiomass, ATM}$ efficiency[a] $((W\ m^{-2})/(\mu g\ m^{-3}))$ |
|---|---|---|
| Local scale dominance (LD) | $10.2 \pm 4.2$ | $10.3 \pm 4.4$ |
| Local scale strong and regional scale weak (LSRW) | $10.6 \pm 5.7$ | $10.2 \pm 5.8$ |
| Local scale weak and regional scale strong (LWRS) | $13.5 \pm 6.7$ | $14.7 \pm 8.1$ |
| Regional scale dominance (RD) | $15.6 \pm 8.9$ | $15.5 \pm 8.4$ |

a: Mean ± Std

**Table R5** The change of mass concentration of different eBCs and their DREs

| Atmospheric motion category | Change of mass concentration of $eBC_{fossil}$ | Change of mass concentration of $eBC_{biomass}$ | Change of $DRE_{eBCfossil, ATM}$ | Change of $DRE_{eBCbiomass, ATM}$ |
|---|---|---|---|---|
| LD | - | - | - | - |
| LSRW | 9.4% | 30.3% | 5.7% | -2.9% |
| LWRS | 30.2% | 43.4% | 29.3% | 23.1% |
| RS | 45.1% | 38.8% | 34.6% | 29.0% |

*Line 32: "It highlights…" what does "it" refer to? In general, this closing sentence reads awkward, and I would suggest rewording it.*

**Response:** Thanks for pointing this out, we have revised the sentence as follows:

"The finding that the DRE efficiency of BC increased during the regional transport suggested significant consequences in regions downwind of pollution sources and emphasizes the importance of regionally transported BC for potential climatic effects."

**Introduction:**

*Light absorbing or agent with positive radiative forcing? The two whings are linked but not the same.*
**Response:** We agree with reviewer. Based on IPCC (2021) and most studies on BC (Zhao, et al., 2019; Panicker et al., 2010; Rajesh and Ramachandran, 2018, Valenzuela et al., 2017) direct radiative forcing, BC generally is considered to be a short life climate forcer which can warm the climate, however for other types of aerosols, their scattering ability is higher than its absorbing ability which more likely leads to negative forcings.

We have revised the sentence into:
"Black carbon (BC) is produced by the incomplete combustion of biomass and fossil fuels. The BC aerosol has a strong light absorption capacity and can cause heating of the atmosphere. In fact, BC is widely recognized as one of the most important short-lived climate forcers (IPCC, 2021)."


*Line 65: "decides" seems more to belong to an intelligent entity. Maybe "determines"*
**Response:** Thanks for explaining the difference between the two words. We have changed the "decides" into

"determines".

"The relationships between atmospheric motions and pollutant concentrations are complex. Atmospheric motions determine where and how extensive the pollution impacts are, but of course the rates of pollutant emissions, especially local ones, are important, too (Dutton, 1976)."

*Line 72: "river valley city" comes a bit out of the blue here, it might be good to provide a sentence with some background, like the general location, etc., even if that's then discussed in detail in the method section. Is the city specifically Baoji?*

**Response:** We agree that providing some background information would make it is more coherent, so we rewrote this part as follows:

"Topography also plays an important role in air pollution (Zhao et al., 2015). River-valley topography is complicated, and it can have a considerable influence on air pollution and synoptic patterns of flow (Green et al., 2016; Carvalho et al., 2006). The pollution levels at cities in river-valleys are not only influenced by general atmospheric dynamics but also strongly impacted by the local-scale of dynamics (Brulfert et al., 2006). Surface albedo and surface roughness are affected by the complex topography of river-valley regions, and those physical factors can affect circulation causing changes in pollutant mass concentrations (Wei et al., 2020). Mountains also significantly affect pollution, and once pollutants are generated or transported into the river-valley regions, their dispersal can be impeded by the blocking effect of the mountains. Instead of being dispersed, they can be carried by the airflows over the mountains to converge at the bottom of the valley and increase the pollutants along the river (Zhao et al., 2015). In this way, pollutants can accumulate in valleys and spread throughout the area, thereby aggravating pollution. In addition, temperature inversions commonly form in river-valleys during the winter, and that, too, can aggravate pollution problems (Glojek et al., 2022 and Bei et al., 2016)."

The river valley city we referred here is not just baoji. This topography (river-valley) has been found impact on air pollution and synoptic patterns in other countries as well (Brulfert et al., 2006, Green et al., 2016, Glojek et al., 2022).


*Line 75: why the albedo makes the solar radiation uneven? Do they mean the reflected radiation? In general, I find this sentence awkward and unclear.*
**Response:** Surface albedo is the ratio of up-welling to down-welling short wave radiative flux at the surface ($Albedo = \frac{Reflected\ radiation}{Incident\ radiation}$). Surface albedo is one of the most important parameters for determining radiative

forcing and it impacts on climatic processes. The spatial and temporal distribution of surface properties captured by albedo reflect a variety of natural and human influences on the surface that are of importance in terms of radiative balance. Surface albedos vary with the type of surface; for example the albedo for the ocean is much lower than that of land (Satheesh et al., 2006), the surface albedo of vegetated lands is also different from that of urban areas. Studies found that lower surface albedos, compared with higher ones, result in more positive radiative forcing at the top of the atmosphere (Nari et al., 2013 and reference therein).

We have rewritten that sentence from line 75 as following and hope it now reads clear and understandable.

"Surface albedo and surface roughness are affected by the complex topography of river-valley regions, and those physical factors can affect circulation causing changes in pollutant mass concentrations (Wei et al., 2020)."

*The sentence starting on line 77 is also awkward and should be reworded.*
**Response:** We have revised this sentence, now it reads like:

"Mountains also significantly affect pollution, and once pollutants are generated or transported into the river-valley regions, their dispersal can be impeded by the blocking effect of the mountains. Instead of being dispersed, they can be carried by the airflows over the mountains to converge at the bottom of the valley and increase the pollutants along the river (Zhao et al., 2015)."

*Line 80: eBC appears here without having been defined as equivalent BC. It has been defined only later, but it should be defined at its first appearance.*
**Response:** Thanks for pointing this out. We have added the definition of eBC here and deleted the one appeared later. Now it is:

"Thus, we focused our study on the impacts of different scales of motion on source-specific equivalent BCs (eBCs), and we evaluated radiative effects of eBCs over a river-valley city."

*Line 81: I believe the authors meant: "the contributions of fossil fuel combustion and biomass burning to eBC concentrations"*
**Response:** Yes, that is what we mean. Thanks for the comment, we have revised this sentence into

"The primary objectives of this study were: (1) to quantify the contributions of fossil fuel combustion and biomass burning to eBC concentrations, (2) to investigate the impacts of different scales of motion on the source-specific eBC, and (3) to estimate the radiative effects and the radiative efficiency of the source-specific eBC under different atmospheric motion scenarios."

**Method:**
*Line 88: Guanzhong Plain is where the river-valley city of Baoji is located, I guess? It would be nice to say so right at the beginning.*
**Response:** We have revised this sentence to make that point clear. The revised paragraph can be seen in a response above to the general comments.

*Section 2.1: a map of the region I think would help as figure 1.*
**Response:** We have updated the map of the region in supplementary materials. The new map looks like:

[Figure]

**Figure R4.** A map of the research site; (a) map of China—the red shape is the location of Baoji, (b) a map of the Guanzhong Plain, the black star represents the location of Baoji; (c) a map of Baoji City, the black dots and the black triangle represent 12 stations and the triangle is the location of sampling site, (d) a map of the sampling site.

*Line 94: dense population, provide some numbers such as the total population, and population density.*
**Response:** We have added the total population (0.341 million with 63.5% population living in downtown aera) and population density (6003 people per km$^2$ in 2019) in the site description. The revised paragraph can be seen in the response above in the general comments.

*Section 2.5 is quite confusing.*
**Response:** We have revised this section and added more detailed explanations of the equation variables and had it edited by a native speaker for clarity. Now the section 2.5 reads:

"**2.5 Indicators for the different scales of motion**
The mathematical definitions of airflow condition proposed by Allwine and Whiteman (1994) were used in this study. The definitions quantify the flow features integrally at individual stations. Three variables were quantified, namely the actual wind run distance ($S$) which is the scalar displacement of the wind in 24 h (i.e. the accumulated distance of the wind), the resultant transport distance ($L$) which is the vector displacement of the wind in 24 h (i.e. the straight line from the starting point to the end point), and the recirculation factor ($R$) is based on the ratio of $L$ and $S$ which indicates the frequency of the wind veering in 24 h. The influences of different scales of atmospheric motions were assessed based on the method proposed by Levy et al., (2010), and for this, we used wind data at 100 m above the sampling site and the wind data from 12 monitoring stations at ground level (~15m) to indicate the different scales of motions. The winds at the surface monitoring stations were expected to be more sensitive to local-scale turbulence and convection than

the winds at 100 m. With less influence from the surface forces, the indicators at 100 m would be more sensitive to larger scales of motion. The equations used as follows:

$$L_{n\tau/bj} = T\left[\left(\sum_{j=i}^{i-\tau+1} u_i\right)^2 + \left(\sum_{j=i}^{i-\tau+1} v_i\right)^2\right]^{1/2} \tag{11}$$

$$S_{n\tau/bj} = \sum_{j=i}^{i-\tau+1}(u_j{}^2 + v_j{}^2)^{1/2} \tag{12}$$

$$R_{n\tau/bj} = 1 - \frac{L_{i\tau}}{S_{i\tau}} \tag{13}$$

where $T$ is the interval of the data (i.e., 60 min), $i$ is the $i^{th}$ the ending time step data, $\tau$ is the integration time period of the wind run (24 h), i-$\tau$+1 represents the data at the start time, and $n$ is the number of monitoring stations (a total of 12 in this study). The quantities $u$ and $v$ are the wind vectors. Using the wind data from the 12 monitoring stations covering Baoji, the $L$ and $S$ values at the 12 different sites at ground level were calculated. $L_{n\tau}$ and $S_{n\tau}$ represent the resultant transport distance and the actual wind run distance at the $n^{th}$ (n = 1 to 12) monitoring station at ground level; $R_{n\tau}$ is the recirculation factor at the $n^{th}$ monitoring station which is calculated based on $L_{n\tau}$; and $S_{n\tau}$; $L_{bj}$, and $S_{bj}$ are the resultant transport distance and the actual wind run distance at 100 m height above the ground. These represent the flow characteristics in higher atmosphere at the study site, and they were calculated by using the wind data at 100 m height. The recirculation factor ($R_{bj}$) was calculated for a height of 100 m.

As explained in Levy et al., (2010), if local-scale motions are strong and regional-scale motions are weak, the variations in winds at each station would not be likely to be uniform due to differences in local factors, and that would result in relatively large standard deviations ($R_{std}$) for $R_{n\tau}$. By contrast, if the local-scale motions are weak and the regional-scale motion is strong, the wind direction would be likely to be more uniform over a large area, and the $R_{bj}$ and the $R_{std}$ should be relatively smaller."

*Line 175: A ratio indicates a difference? That is confusing. Also, R is defined in 13 not as the ratio of L and S but as the ratio of the difference between S and L and S itself. Or is this a different R?*
**Response:** We apologize for the inaccurate expression. The "R" on line 175 is the R defined in equation 13. We have corrected the description of the R in line 175. Now it consistent with the equation 13:

"Three variables were quantified, namely the actual wind run distance ($S$) which is the scalar displacement of the wind in 24 h (i.e. the accumulated distance of the wind), the resultant transport distance ($L$) which is the vector displacement of the wind in 24 h (i.e. the straight line from the starting point to the end point), and the recirculation factor ($R$) is based on the ratio of $L$ and $S$ which indicates the frequency of the wind veering in 24 h."

*Line 189: Again, if equation 13 is correct, then R is not the ratio of L to S but 1 – the ratio of L to S. Same for line 191 (which seems a repetition anyway).*
**Response:** As noted above, we have corrected the expression that was on line 189 and 190, now it reads like:

"$R_{n\tau}$ is the recirculation factor at the $n^{th}$ monitoring station which is calculated based on $L_{n\tau}$ and $S_{n\tau}$; $L_{bj}$, and $S_{bj}$ are the resultant transport distance and the actual wind run distance at 100 m height above the ground. These represent the flow characteristics in higher atmosphere at the study site, and they were calculated by using the wind data at 100 m height. The recirculation factor ($R_{bj}$) was calculated for a height of 100 m."

*Lines 192-193: this seems a bit of a circular argument (a tautology).*
**Response:** We have rewritten this paragraph as follows:

"As explained in Levy et al., (2010), if local-scale motions are strong and regional-scale motions are weak, the variations in winds at each station would not be likely to be uniform due to differences in local factors, and that would result in relatively large standard deviations ($R_{std}$) for $R_{nr}$. By contrast, if the local-scale motions are weak and the regional-scale motions are strong, the wind direction would be likely to be more uniform over a large area, and the $R_{bj}$ and the $R_{std}$ should be relatively smaller."

*Line 207: provide a citation.*
**Response:** We have provided the citation as following:

"Determining the size of the output map is crucial for SOM (Chang et al 2020 and Liu et al., 2021)."

**Results and discussion:**
*Line 261: contain -> include*
**Response:** Thanks for the correction, we have changed the word:

"The important input data included aerosol parameters, including aerosol optical depth (AOD), single scattering albedo (SSA), asymmetric factor (AF) and extinction efficiency, surface albedo, and atmospheric profile."

*Line 222: what is the default ratio in the model? And what model? OPAC?*
**Response:** Yes, the model is OPAC. The default ratio used for converting mass concentration of BC to its number concentration is 5.99E-5 ($\mu$g m$^{-3}$/ part.cm$^{-3}$) in OPAC. To make this clear, we have added this information into the sentence to make it clear:

"The number concentrations of soot were derived from the mass concentrations of eBC with the default ratio (5.99E-5 $\mu$g m$^{-3}$/ particle.cm$^{-3}$) in OPAC."

*Line 230: at -> of?*
**Response:** We corrected this.

"where $DRE_{eBC}$ is the DRE of source-specific eBC, $F\downarrow$ and $F\uparrow$ are the downward and upward flux, $DRE_{eBC,ATM}$ is the DRE of the source-specific eBC for the atmospheric column, that is, the DRE at the top of the atmosphere ($DRE_{eBC,TOA}$) minus that at the surface ($DRE_{eBC,SUF}$)."

*Lines 238-239: Provide some more background or at least some references*
**Response:** Thanks for the suggestion. We have added some background information about the Bootstrap (BS) and Displacement (DISP) with some references. Now it reads:

"Two diagnostic methods, Bootstrap (BS) and Displacement (DISP) (Norris et al, 2014; Brown et al. 2015) were used to validate the robustness and stability of the results. The BS method was used to assess the random errors and partially assess the effects of rotational ambiguity while DISP was used to evaluate rotational ambiguity errors. The results of the BS and DISP analyses showed that there was no swap for the 4-factor solution (Table S3)."

*Line283: variations… varied… rephrase.*
**Response:** We have corrected this awkward sentence:

"The results showed that eBC_fossil and eBC_biomass were only weakly correlated (r = 0.3, Figure S9), indicating a reasonably good separation, and furthermore, their diel variations showed different patterns (Figure 2)."

*Lines 293 – 294. Remove "New para here"*
**Response:** We deleted it.

*Line 295: why did the biomass burning increase after 6 pm? Is it indoor biomass burning for cooking or heating, or is it some other biomass burning?*
**Response:** Based on previous research (Xie et al., 2010, Zhou et al., 2018) and what we know about residential energy usage in rural areas of Baoji, biomass is commonly used as fuel for residential cooking and heating. Therefore we believe that the biomass burning increased after 6 pm due to the evening meal preparation and residential heating.

To make this clear, we have revised this:

"In contrast, the diel variation of eBC_biomass (Figure 2b) showed greater influences from meteorological conditions during the daytime, and eBC_biomass showed lower concentrations during the day compared with the night. After 6 p.m., increased biomass burning from cooking and residential heating let to the emission of more eBC_biomass and the stable PBLH hindered the dispersion of eBC_biomass; these two factors caused the eBC_biomass to reach its peak at 8 p.m."

*Figure 3, caption: Explain the meaning of the gray and yellow areas even if that's explained in the text.*
**Response:** We added an explanation to the figure caption as follows:

[Figure]

**Figure R5.** The $75^{th}$ – $100^{th}$ percentile mass concentrations of the eBC from fossil fuel combustion (eBC_fossil) and (b) the eBC from biomass burning (eBC_biomass) under local scale dominance (LD, red circle), local scale strong and regional scale weak (LSRW, green circle), local scale weak regional scale strong (LWRS, purple circle) and regional scale dominance (RD, blue circle). $S_{bj}$ is actual wind run distance at 100m height, $R_{bj}$ is the recirculation factor, the grey area indicates good ventilation ($S_{bj} \geq 250km$, $R_{bj} \leq 0.2$), the yellow area indicates air stagnation ($S_{bj} \leq 130km$).

*Line 344: can one try to verify this with the backtrajectory analysis, or satellite products. etc.*
**Response:** To verify the sources of eBC$_{biomass}$ located further than that of eBC$_{fossil}$, we used non-parametric wind regression plots (Gu et al, 2020). As shown in the Figure R6, during the night, the eBC$_{biomass}$ was higher when the wind speed was higher, while the eBC$_{fossil}$ was higher when the winds were lower. This indicates that during night, the emission sources of eBC$_{biomass}$ were located further than the sources for eBC$_{fossil}$, which explains why the mass concentrations of eBC$_{biomass}$ did not decrease with the influence of regional-scale atmospheric motion increased. We have added this figure into the supplementary materials.

[Figure]

**Figure R6** Non-parametric wind regression plots for eBC$_{biomass}$ (a) and eBC$_{fossil}$ (b) at night. The radial and tangential axes represent the wind direction (°) and speed (m s$^{-1}$), respectively, nws100m represents the night wind speed 100m above the ground level.

**Response:** Yes, we agree. The higher DRE efficiencies can be attributed to the enhanced MAC of BC during the regional transport. So we have added the following text into the paragraph:

"Although $DRE_{eBC, ATM}$ declined with increased influences from the regional scale of motion, the $DRE_{eBC, ATM}$ efficiency ($DRE_{eBC, ATM}$ per mass concentration) was found to increase with greater regional-scale motion. Furthermore, the DRE efficiencies of both types of eBC under LD and LSRW were comparable, around 10 W m$^{-2}$ (Table 2). In contrast, the efficiencies varied more when the regional-scale motions were stronger. Under LWRS, the efficiencies of $eBC_{fossil}$ and $eBC_{biomass}$ were 13.5 ± 6.7 and 14.7 ± 8.1 (W m$^{-2}$)/(μg m$^{-3}$) respectively. Under RD, the efficiencies were even higher, 15.6 ± 8.9 (W m$^{-2}$)/(μg m$^{-3}$) for $eBC_{fossil}$ and 15.5 ± 8.4 (W m$^{-2}$)/(μg m$^{-3}$) for $eBC_{biomass}$, which are > 1.5 times those recorded under LD. The higher eBC efficiencies may have been caused by the increases in the BC MAC during the regional transport. Studies have confirmed that the aging processes in the atmosphere can enhance the light-absorbing ability of BC (Chen et al., 2017; Shen et al., 2014), and regional transport can provide sufficient time for BC aging (Shiraiwa, et al. 2007; Cho et al., 2021)."

*Caption of figure 5: explain what the different shadings represent. Also, x-axis label number four probably should be "DREeBCfossil, TOA" not "DREeBCbiomass, TOA". It could be interesting to add another two panels with the equivalent calculations but in terms of efficiency to make more clear what is discussed in words in the paper.*
**Response:** Thank you for pointing out this mistake. We have corrected the label and explained the shading and as suggested, we also added two figures. The figure has been replaced with following one:

[Figure]

**Figure R7.** Direct radiative effect (DRE) of the eBC from fossil fuel combustion (eBC$_{fossil}$) shaded in grey and the eBC from biomass burning (eBC$_{biomass}$) shaded in yellow (a) in the top atmosphere (TOA), surface (SUF), and the atmosphere atmospheric column (ATM) and (b) the DRE$_{eBC,ATM}$ of two types of eBC under local scale dominance (LD) shaded in light grey labeled as LD, local scale strong and regional scale weak (LSRW) shaded in light blue labeled as LSRW, local scale weak regional scale strong (LWRS) shaded in light grey labeled with LWRS and regional scale dominance (RD) shaded in light blue labeled as RD (c) DRE efficiencies of eBC$_{biomass}$ (shaded in yellow) and eBC$_{fossil}$ (shaded by grey) in TOA, SUF and ATM (d) DRE efficiencies of eBC$_{biomass}$ and eBC$_{fossil}$ at ATM under LD (shaded in light grey labeled as LD), LSRW (shaded in light blue labeled as LSRW), LWRS (shaded in light grey labeled as LWRS) and RD (shaded in light blue labeled with RD).

---

## Author Comment (AC2)

*Comment on acp-2022-26 titled "Review of "The impact of atmospheric motion on source-specific black carbon and the induced direct radiative effect over a river-valley region" by Liu et al.*

**Anonymous Referee #1**

**General comment**

*This manuscript describes how Black Carbon concentrations, measured over a little more than a month in winter 2018 together with other chemical species in $PM_{2.5}$ in a Chinese city nestled at the bottom of a valley, vary with different transport regimes and motion scales (from local to regional).*

**Response:** We sincerely thank the reviewer for the comments and suggestions, and we have extensively revised the relevant text and modified the content. Below are point-by-point responses—the modifications to the manuscript are included.

*The topic of the manuscript fits within the scope of the journal. The methodology should be described more precisely to build confidence in the results.*

**Response:** We revised the methods to provide more details, particularly for the optical source apportionment and SOM. Other methods also have been revised according to the reviewer's suggestions. In addition, we have provided information regarding the cluster analysis and minimum R squared method in the supplementary materials. The changes are shown in the revised version. The revised paragraphs now read as follows:

[revised manuscript text omitted]

**"Text S1. Minimum R‑squared method**

The minimum R squared method developed by Wu et al., (2016) was used to separate secondary organic carbon (SOC) from the primary organic carbon (POC). The assumption behind this method is the organic carbon (OC) from non-combustion source is negligible. As explained by Wang et al., (2019), the major non-combustion source is biogenic which is mainly exists in coarse mode. Thus, the non-combustion organic carbon is considered negligible in this study. Therefore, SOC and POC can be separated by using following equations. For each date set, the ratios of OC to eBC and SOC and the $R^2$ between eBC and SOC can be calculated. SOC and eBC are considered independent, so the (OC/eBC)pri should be the value obtained when the $R^2$ between eBC and SOC is minimum.

$$POC = (OC/EC)_{pri} \times EC \tag{S1}$$

$$SOC = OC_{total} - (OC/EC)_{pri} \times EC \tag{S2}$$

where EC in this study is eBC. The (OC/EC)pri is the ratio in freshly emitted OC and EC from combustion sources.

The light absorption at shorter wavelengths (<660nm) is not only from primary light absorbing substances but also from the secondary organic carbon (Wang et al., 2019). The assumption for this method is that the light absorption caused by non-combustion sources is negligible. As mentioned above, most of the biogenic BrC is in coarse mode. Another common light absorbing substance is the $Fe_2O_3$ in the dust, but the impact of that should be limited because the absorption from $Fe_2O_3$ in the dust has been reported to be much smaller than that from BC (Ramachandran and Kedia, 2010). Thus, to separate the secondary light absorption ($b_{abs}(\lambda)_{secondary}$) from the primary light absorption ($b_{abs}(\lambda)_{primary}$), a BC-tracer method coupled with a minimum R-squared method was used. The equations used for the calculation are follows:

$$b_{abs}(\lambda)_{secondary} = b_{abs}(\lambda) - (\frac{b_{abs}(\lambda)}{BC})_{pri} \times BC \tag{S3}$$

$$b_{abs}(\lambda)_{primary} = b_{abs}(\lambda) - b_{abs}(\lambda)_{secondary} \tag{S4}$$

Where $b_{abs}(\lambda)$ is the light absorption at different wavelengths ($\lambda$=370nm, 470nm, 520nm, 590nm, 660nm) measured by AE33, BC is the eBC measured by AE33 at a wavelength of 880nm. The $(\frac{b_{abs}(\lambda)}{BC})_{pri}$ is the ratio of the primary light absorption to the BC mas concentration from combustion sources.

**Text S2. Cluster analysis of air-mass trajectories**

Back trajectories were calculated by using Hybrid Single-Particle Lagrangian Integrated Trajectory (HYSPLIT) model (Draxler and Hess, 1998) developed by the Air Resource Lab (ARL) of the National Oceanic and Atmospheric Administration (NOAA). The model can predict the position of air mass by using mean wind. The back-in-time positions are calculated by reversing the advection equation (Draxler and Hess, 1997). The calculation requires the mean wind, for calculating trajectories, only advection is considered (Stein et al., 2015). The basic equations for trajectory calculation in HYSPLIT are as follows:

$$P'(t + \Delta t) = P(t) + V(P, t) \times \Delta t \qquad (S5)$$

$$P(t + \Delta t) = P(t) + 0.5 \times [V(P, t) + V(P', t + \Delta t)] \times \Delta t \qquad (S6)$$

Where $P(t)$ is the initial position, $P'(t + \Delta t)$ is the first guess position, $V$ is the average velocity, $t$ is the time, $\Delta t$ is the time step.

A large number of 24 h trajectories (793) that were retrieved for the study period showed diverse pathways, so in order to find out the representative pathways for those trajectories, a cluster analysis based on an angle-based distance statistics method was conducted. Compared with Euclidean distance, angle-based distance statistics method focuses on the direction of air mass instead of the speed. The angle-based distance statistics method is defined by following equations (Sirois and Bottenheim, 1995):

$$d_{12} = \frac{1}{2} \sum_{i=1}^{n} \cos^{-1}(0.5 \times \frac{A_i + B_i - C_i}{\sqrt{A_i B_i}}) \qquad (S7)$$

$$A_i = (X_1(i) - X_0)^2 + (Y_1(i) - Y_0)^2 \qquad (S8)$$

$$B_i = (X_2(i) - X_0)^2 + (Y_2(i) - Y_0)^2 \qquad (S9)$$

$$C_i = (X_2(i) - X_1(i))^2 + (Y_2(i) - Y_1(i))^2 \qquad (S10)$$

Where $d_{12}$ is the average angle between the two backward trajectories, varying between 0 and $\pi$; $X_0$ and $Y_0$ are the position of the receptor site; and $X_1$ ($Y_1$) and $X_2$ ($Y_2$) are the backward trajectories 1 and 2, respectively. In this study, three clusters were chosen as representative of the backward trajectory clusters based on the total spatial variance (TSV) value. The simulation was conducted using the GIS-based TrajStat software (Wang et al., 2009)."

*The discussion section could be strengthened by comparing this work with other studies in similar topographic conditions.*

**Response:** We have reviewed studies on BC pollution and the total DRE of BC at other river valley sites and compared them with this study. The revised text is as follows:

"The mean values of eBC$_{fossil}$ and eBC$_{biomass}$ were 2.46 µg m$^{-3}$ and 1.17 µg m$^{-3}$, respectively. The averaged total eBC mass concentration (± standard deviation) was 3.63±2.73µg m$^{-3}$, and the eBC ranged from varying from 0.39 to 12.73 µg m$^{-3}$ during the study period, The averaged mass concentration was comparable to that in Lanzhou, another river valley city in China, that was sampled in the same season (5.1 ± 2.1, Zhao et al.,2019). The lowest value is comparable to other river valley regions such as in Retje in India (Glojek et al.,

2022) or in Urumqi River Valley in China (Zhang et al., 2020), however even the highest concentration was much lower than that in other urban regions (Table S5)."

"Figure 5a shows the DREs at top of the atmosphere (DRE$_{eBC, TOA}$), surface (DRE$_{eBC, SUF}$), and the whole atmosphere (DRE$_{eBC, ATM}$) of eBC$_{fossil}$ and eBC$_{biomass}$. The DRE$_{eBC, TOA}$ and DRE$_{eBC, SUF}$ of eBC were 13 W m$^{-2}$ and -22.9 W m$^{-2}$, which were lower than that reported in Lanzhou (21.8 W m$^{-2}$ and -47.5 W m$^{-2}$ for DRE$_{eBC, TOA}$ and DRE$_{eBC, SUF}$) – which is another a river valley city in China (Zhao et al., 2019). This could be due to fact that the eBC mass concentration in Baoji was lower than in Lanzhou (Table S5). As for the DRE$_{eBC, TOA}$ and DRE$_{eBC, SUF}$ per an unit mass of BC, the results of the two studies were comparable."

**Table R1** Mean (range) BC mass concentration in river valley sites worldwide

| Reference | BC concentration ($\mu$g m$^{-3}$) | Season | Topographic conditions | Altitude | Station type | Year |
|---|---|---|---|---|---|---|
| This study | 3.63±2.73 (0.39~12.73) | November~December (winter) | river valley | 450 to 800 m a.s.l.[a] | urban | 2018 |
| Glojek et al., (2020) | 0.9~40 | December~January (winter) | river valley | 715 m a.s.l. | rural | 2017-2018 |
| Zhao et al. (2015) | 25±11 | January (winter) | river valley | 410 m a.s.l. | urban | 2013 |
| Barman and Gokhale (2019) | 20.58~22.44 | Winter | river valley | | urban | 2016-2017 |
| Zhang et al., 2020 | 0.102~1.525 | Winter | river valley | 2130 m a.s.l. | rural | 2016-2017 |
| Chakrabarty et al., (2012) | 9~41 | January~February (winter) | river valley | | urban | 2011 |
| Zhao et al. (2019) | 5.1 ± 2.1 | December~January (winter) | river valley | | urban | 2018 |
| Tiwari et al., 2016 | 8.19 ±1.39 | December-February (winter) | river valley | 55 m a.s.l. | urban | 2013-2014 |

[a]asl stands for "above sea level."


*The writing should be improved, I have suggested a few technical corrections in a specific section at the end.*

**Response:** As suggested, we have had this manuscript polished by a native English speaker.

**Specific comments:**

*L30: What type of "change" between a mass concentration and a radiative effect is expected? I am not sure the word "change" conveys your meaning.*

**Response:** By "change" we meant that the changes in BC mass concentrations for the different scales of motions were not of the same magnitude as the changes in BC DRE. More specifically, we took LD as a base case to calculate the difference between the average BC mass concentrations (or average DRE) versus other cases (LSTW, LWRS and RD). Clearly the $eBC_{fossil}$ and $eBC_{biomass}$ concentrations decreased more than the corresponding DRE did (Table R2). This indicates that the DREs per unit mass of BC were variable (Table R3).

To avoid confusion, we revised it into "Similar to the mass concentrations, the DREs of the two types of eBC were both lower when the regional scale of motions were greater than the local ones. However, the changes in mass concentrations and DREs were not proportionate because the regional-scale of motions carried the fresh BC away from the local site but brought the aged BCs to the site from the upwind regions. As a result, the DRE efficiency of eBC was ~1.5 times higher when the regional scale of motion was stronger."

**Table R2** The change of mass concentration of different eBCs and their DREs

| Atmospheric motion category | Change of mass concentration of eBCfossil | Change of mass concentration of $eBC_{biomass}$ | Change of $DRE_{eBCfossil, ATM}$ | Change of $DRE_{eBCbiomass, ATM}$ |
|---|---|---|---|---|
| LD | - | - | - | - |
| LSRW | 9.4% | 30.3% | 5.7% | -2.9% |
| LWRS | 30.2% | 43.4% | 29.3% | 23.1% |
| RD | 45.1% | 38.8% | 34.6% | 29.0% |

**Table R3.** Direct radiative forcing efficiencies for equivalent black carbon (eBC) from fossil fuel combustion ($eBC_{fossil}$) and the eBC from biomass burning ($eBC_{biomass}$) under four atmospheric motion categories

| Atmospheric motion category | DRE$_{eBCfossil, ATM}$ efficiency ((W m$^{-2}$)/(μg m$^{-3}$)) | DRE$_{eBCbiomass, ATM}$ efficiency ((W m$^{-2}$)/(μg m$^{-3}$)) |
|---|---|---|
| Local scale dominance (LD) | 10.2[a] ± 4.2[b] | 10.3 ± 4.4 |
| Local scale strong and regional scale weak (LSRW) | 10.6 ±5.7 | 10.2 ± 5.8 |
| Local scale weak and regional scale strong (LWRS) | 13.5 ± 6.7 | 14.7 ± 8.1 |
| Regional scale dominance (RD) | 15.6 ± 8.9 | 15.5 ± 8.4 |

a and b: Mean ± Standard deviation

*L37-38: "the second strongest light-absorbing substance in the atmosphere after CO2". This wording is confusing since it seems an intrinsic property of BC whatever its concentration level. Besides it is clearly related to (1) its climate forcing ability and (2) human emissions only, by Bond et al. (2013) which state that "We estimate that black carbon, with a total climate forcing of +1.1 W m$^{-2}$, is the second most important human emission in terms of its climate forcing in the present-day atmosphere; only carbon dioxide is estimated to have a greater forcing." (abstract) or "Our best estimate of black carbon forcing ranks it as the second most important individual climate-warming agent after carbon dioxide" (in 1.2.12 Policy implications). This sentence should thus be revised accordingly.*

**Response:** Thanks for pointing this out, we have revised this sentence:

"Black carbon (BC) is produced by the incomplete combustion of biomass and fossil fuels. The BC aerosol has a strong light absorption capacity and can cause heating of the atmosphere. In fact, BC is widely recognized as one of the most important short-lived climate forcers (IPCC, 2021)."

In the abstract, the relevant sentence has also been revised into:

"Black carbon (BC) is one of the most important short lived climate forcers, and atmospheric motions play an important role in determining its mass concentrations of pollutants."

*L60: "scale (it is atmosphere phenomena) ranges". I do not understand what the mention in brackets refers to?!*

**Response:** We have deleted this—it was a mistake.

*L62: I do not understand how an atmospheric dynamic feature (the local scale of motion) is eventually controlled by the concentration levels of BC?! Besides the land roughness, it can be influenced by thermals, turbulence, etc.*

**Response:** We agree with reviewer that thermals and other factors cause local scales of motion, but the variations in the BC mass concentrations also can induce thermal differences horizontally and vertically. The horizontal temperature variations can give rise to horizontal pressure differences, which result in atmospheric motions (Oke, 1988). BC concentration also can impact the temperature structure of the atmosphere and cloud microphysical properties, the latter of which also can impact the temperature (IPCC, 2021). In this way, differences of BC mass concentrations between different locations can lead to atmospheric motions.


**Table R4.** The seasonal meteorological data of Baoji

| Season | Temperature (°C) | Relative humidity (%) | Precipitation in last hour (mm) |
|--------|------------------|------------------------|----------------------------------|
| Winter | 2.7 | 60.6 | 0.025 |
| Spring | 11.5 | 54.9 | 0.042 |
| Summer | 23.7 | 67.1 | 0.139 |
| Autumn | 20.2 | 67.0 | 0.074 |

[Figure]

**Figure R1** Seasonal wind roses for Baoji.

We also have updated the map of the region in supplementary materials. The new map looks like:

[Figure]

**Figure R2.** A map of the research site; (a) map of China—the red shape is the location of Baoji, (b) a map of the Guanzhong Plain, the black star represents the location of Baoji; (c) a map of Baoji City, the black dots and the black triangle represent 12 stations and the triangle is the location of sampling site, (d) a map of the sampling site.

*L101: The time resolution of AE33 aethalometers can either be 1 min or 1 s. Do you mean you used 5-min averages of the 1-min data?*

**Response:** The original data had 1-min time resolution. This has been corrected:

"eBC and the absorption coefficients ($b_{abs}$) at 370, 470, 520, 590, 660, 880, and 950 nm wavelength were measured using an AE33 aethalometer (Magee Scientific, Berkeley, CA, USA) equipped with a $PM_{2.5}$ cut-off inlet (SCC 1.829, BGI Inc. USA) that had a time resolution of 1 min."

*L105-106: What type of quartz filter was used? The correction factor is dependent on the filter model.*

**Response:** We double checked with the one of the coauthors about the tape and the C value used for this sampling. The tape is PN8060. We reset the C value to 1.39 instead of 2.14. Thanks so much for spotting this omission. We corrected this in the revised version.

"The quartz filter (PN8060) matrix scattering effect was corrected by using a factor of 1.39."

*L133: What is the spatial resolution of the GDAS meteorological data?*

**Response:** The spatial resolution of GDAS data is 1°×1°. We have added the resolution into the manuscript and provide a table with data and the main parameters used for the HYSPLIT model (Table R5).

"The data used for the Hybrid Single-Particle Lagrangian Integrated Trajectory (HYSPLIT) model was downloaded from Global Data Assimilation System and it had a resolution of 1°×1° (GDAS, https://www.ready.noaa.gov/gdas1.php). The data and main parameters used in trajectory model are listed in Table S2."

**Table R5.** Data and parameters used in HYSPLIT model

| Items | Data/parameters |
|---|---|
| Model | HYSPLIT |
| Meteorological data | GDAS data, 1° × 1°, 23 vertical levels, 3 hourly |
| Backward period | 24h |
| Footprint level | 100 m above the ground |
| Receptor site location | 34°21′16.8″N, 107°12′59.6″E |

*L162: The PMF methodology is described in section 2.3 for its classical use with mass concentrations of chemical species. Here it is used for optical source apportionment and therefore it should be explained how (concentrations are being replaced by what?) as well as the uncertainty calculation in that case.*

**Response:** The use of the input data follows equation (R1)

$$X_{ij} = \sum_{k=1}^{p} g_{ik}f_{kj} + e_{ij} \tag{R1}$$

where $X_{ij}$ denotes the input data matrix; $p$ is the number of sources selected in the model; $g_{ik}$ denotes the contribution of the $k^{th}$ factor to the $i^{th}$ input data; $f_{kj}$ represents the $k^{th}$ factor's profile of the $j^{th}$ species; and $e_{ij}$ represents the residual. Both $g_{ik}$ and $f_{kj}$ are non-negative.

As shown in the equation R1, the balance (mass balance for chemical data, total absorption balance for absorption data) isn't be violated by using a mixed input matrix X because each $X_{ij}$ is independent from the others. PMF analysis is not degraded a priori by using joint matrices containing different dimensions/units (Forello et al., 2019 and reference therein). Putting data with different units in X means that matrix G ($g_{ik}$) is unitless and the corresponding column data in F ($f_k$) will have the same unit as the total values you into X (Paatero, 2018).

In addition, we were not interested in the total particulate mass which would satisfy the condition that the mass is equal to the sum of the mass of each chemical species. Instead, we focused only on the relationship between EC and light absorption coefficients to retrieve AAE values. We used chemical species only as tracers for different emission sources to help identify the factors. Thus, we simply use a joint matrix X[Y Z] where Y represents the part of the matrix for chemical species, Z represents the part of the matrix for light absorption coefficients.

The uncertainty of the input data was calculated by the following equation recommended in EPA PMF5.0 user guideline:

$$\text{Unc} = \sqrt{(\text{Error Fraction} \times \text{concentration})^2 + (0.5 \times \text{MDL})^2} \tag{R2}$$

$$\text{Unc} = \frac{5}{6} \times \text{MDL} \tag{R3}$$

where MDL is the minimum detection limit of different input data. When the concentration of species was higher than MDL then equation R2 was used otherwise equation R3 was used.

The error fraction for the offline measured data was the analytical uncertainty which in this study was taken to be the difference between replicate analyses of the same sample. The MDL was used based on the MDL of each chemical species as measured by the analytical instrumentation. For light absorption coefficients, the error fraction used was the measurement uncertainty of AE33. The uncertainty was estimated about 10% including changes in filter scattering caused due to aerosol loading, underestimation of source signals with increased filter loading, sample flow rate, filter spot area, and detector response (Rajesh and Ramachandran, 2019). Thus, we used 10% for light absorption coefficients for all wavelengths. The MDL of $b_{abs}$ we used the value of 0.039M m$^{-1}$ which is converted from the reported BC MDL of AE33 (0.005μg m$^{-3}$) in Rajesh and Ramachandran (2018).

To make the calculation clear, we have revised section 2.3, which can be seen in the response above.

*L184: Is equation 13 correct? Both L175 and L188-189 describe R as the ratio between L and S but this does not correspond to the expression.*

**Response:** We apologize for the inaccurate expression. The "R" on line 175 is the R defined in equation 13. We have corrected the description of the R in line 175. Now it consistent with the equation 13:

"Three variables were quantified, namely the actual wind run distance ($S$) which is the scalar displacement of the wind in 24 h (i.e. the accumulated distance of the wind), the resultant transport distance ($L$) which is the vector displacement of the wind in 24 h (i.e. the straight line from the starting point to the end point), and the recirculation factor ($R$) is based on the ratio of $L$ and $S$ which indicates the frequency of the wind veering in 24 h."

*L206: What is the time resolution of the three sets of data? Are you using hourly averages?*

**Response:** Yes, we used hourly averaged data. To make this clear, we added the time resolution into the sentence:

"Hourly averages of three sets of data ($R_{std}$, $L_{bj}$, and $S_{bj}$) were input into SOM. Determining the size of the output map is crucial for SOM (Chang et al 2020 and Liu et al., 2021)."

*L303-318: I would suggest to include the four different average values (L(bj), S(bj), R(bj) and R(std)) for the four motion categories in Table 1 as well, and not repeat all the values in this paragraph but rather focus on the comparison and the interpretation.*

**Response:** Thanks for the suggestion, we have revised this part in 3.2 section to make this section more concise and readable. The revised version is:

"The K-means results showed that the four-category solution was appropriate for interpretation as explained above (see also Figure S10). Thus a 2×2 map size was used for the self organizing map (SOM). The four featured atmospheric motion categories given by SOM (Figure S11) were identified as follows (feature values are in Table 1):

1. Local-scale dominance (LD): This category featured high $R_{bj}$ and $R_{std}$. As described in section 2.5, high $R_{std}$ indicates greater divergence of $R$ at the 12 stations due to the strong influence of local-scale turbulence and convection. $L_{bj}$ and $S_{bj}$ were shorter than 130 km implying stagnation (Allwine and Whiteman, 1994).

2. Local-scale strong and regional-scale weak (LSRW): For this group, $L_{bj}$ and $S_{bj}$ were longer than those for LD*,* and $R_{std}$ was slightly lower than that in LD.

3. Local-scale weak and regional-scale strong (LWRS): As the values suggest, both $R_{bj}$ and $R_{std}$ were lower than those in LD and LSRW, especially $R_{bj}$. This suggests the winds veered less frequently and the differences of $R$ found in 12 stations were smaller than in the two situations above. This situation shows that the influence of the regional-scale motion was greater than that for the previous two categories.

4. Regional-scale dominance (RD): In this category, wind direction at the study site was nearly uniform (extremely low $R_{bj}$) suggesting good ventilation (Allwine and Whiteman, 1994). The differences among $R$ found at the 12 stations were even smaller than for the LWRS group, implying a strong increased influence of regional-scale motions. Indeed, the influence of regional-scale motions far outweighed the

local ones for this category, and therefore, this group was considered to be dominated by strong regional-scale motions."

**Table R6.** The mass concentration of eBC from fossil fuel combustion (eBC$_{fossil}$) and eBC from biomass burning (eBC$_{biomass}$) associated with different clusters under four featured atmospheric motions

| Motion category | Local scale dominance (LD) (40%) | | | | Local scale strong and regional scale weak (LSRW) (17%) | | | |
|---|---|---|---|---|---|---|---|---|
| | $L_{bj}$ = 70.9 km, $S_{bj}$ = 107.8 km, $R_{bj}$ = 0.35, $R_{std}$ = 0.25 | | | | $L_{bj}$ = 106.9 km, $S_{bj}$ = 164.8 km, $R_{bj}$=0.33, $R_{std}$ = 0.23 | | | |
| | Cluster 1 | Cluster 2 | Cluster 3 | Total average | Cluster 1 | Cluster 2 | Cluster 3 | Total average |
| Trajectory percentage (%) | 45 | 52 | 3 | 100 | 56 | 33 | 11 | 100 |
| eBC$_{fossil}$ ($\mu g\ m^{-3}$) | 2.82[a] ± 1.59[b] | 3.2 ± 1.73 | 3.64 ± 0.67 | 3.08 ± 2.07 | 2.42 ± 1.00 | 3.43 ± 1.17 | 2.89 ± 1.00 | 2.79 ± 1.73 |
| eBC$_{biomass}$($\mu g\ m^{-3}$) | 1.34 ± 1.07 | 1.72 ± 1.29 | 0.67 ± 0.87 | 1.52 ± 1.19 | 1.0 ± 0.85 | 1.17 ± 0.84 | 1.00 ± 0.64 | 1.06 ± 0.83 |

$L_{bj}$—resultant transport distance, $S_{bj}$—actual wind run distance at 100 m, $R_{bj}$ —recirculation factor at 100 m, $R_{std}$—standard deviation for recirculation factor. a and b: Mean ± Standard deviation.

| Motion category | Local scale weak and regional scale strong (LWRS) (14%) | | | | Regional scale dominance (RD) (29%) | | | |
|---|---|---|---|---|---|---|---|---|
| | $L_{bj}$ =159 km, $S_{bj}$ = 183.4 km, $R_{bj}$=0.13, $R_{std}$ = 0.20 | | | | $L_{bj}$ =235.6 km, $S_{bj}$ = 246.4 km, $R_{bj}$= 0.05, $R_{std}$ = 0.18 | | | |
| | Cluster 1 | Cluster 2 | Cluster 3 | Total average | Cluster 1 | Cluster 2 | Cluster 3 | Total average |
| Trajectory percentage (%) | 42 | 22 | 36 | 100 | 41 | 20 | 39 | 100 |
| eBC$_{fossil}$ ($\mu g\ m^{-3}$) | 1.32[a] ± 0.67[b] | 2.02 ± 0.73 | 3.16 ± 1.19 | 2.15 ± 1.62 | 1.00 ± 0.64 | 1.02 ± 0.88 | 2.75 ± 1.26 | 1.69 ± 1.36 |
| eBC$_{biomass}$($\mu g\ m^{-3}$) | 0.67 ± 0.49 | 0.73 ± 0.47 | 1.19 ± 0.60 | 0.86 ± 0.58 | 0.64 ± 0.63 | 0.87 ± 0.69 | 1.26 ± 0.68 | 0.93 ± 0.72 |

$L_{bj}$—resultant transport distance, $S_{bj}$—actual wind run distance at 100 m, $R_{bj}$ —recirculation factor at 100 m, $R_{std}$—standard deviation for recirculation factor. a and b: Mean ± Standard deviation.

*L341-344: Are emission inventories available to support the assumption that eBC from biomass burning is more regional then eBC from fossil fuel combustion? L388: A similar conclusion as above is reached here using the back trajectory cluster analysis.*

**Response:** We thank reviewer for this suggestion. We considered using emission inventories to investigate this, however to our knowledge, the available emission inventory is Multi-resolution Emission Inventory for China (MEIC) which contains 6 emission categories (stationary combustion, industrial processes, mobile source, agriculture source, and residential source). We currently do not have access to a detailed source section (e.g. the individual biomass burning source and fossil fuel combustion source) inventory, which is what would be most helpful. Thus, it is not possible to show the source of biomass burning and fossil fuel combustion in an inventory map.

*Non-parametric wind regression plots on these two variables could also be informative to assess the local vs. regional influence. See for instance: Gu et al. https://www.sciencedirect.com/science/article/pii/S0160412019342369 or Pandey et al.: https://link.springer.com/article/10.1007/s10661-022-09879-9*

**Response:** As suggested, we have drawn the non-parametric wind regression plots for eBC$_{biomass}$ and eBC$_{fossil}$. As shown in Figure R3, during the night, the eBC$_{biomass}$ was higher when the wind speed was faster, but in contrast, eBC$_{fossil}$ was higher when the wind was slower. This indicates that during night, the emission sources of eBC$_{biomass}$ were located further than the sources of eBC$_{fossil}$, which explains why the mass concentrations of eBC$_{biomass}$ did not decrease when the influence of regional-scale atmospheric motions was stronger. We have added this figure to the supplementary materials.

[Figure]

**Figure R3** Non-parametric wind regression plots for eBC$_{biomass}$ (a) and eBC$_{fossil}$ (b) at night. The radial and tangential axes represent the wind direction (°) and speed (m s$^{-1}$), respectively, and nws100 m represents the night wind speed 100 m above ground level.

**Supplementary Information**

*Figure S5: eBC(diesel) is actually EC from the PMF factor attributed to diesel vehicular emissions if I understand correctly what is written in the main text (L248). I do not think it should be considered equivalent to an eBC(diesel) which is confusing with eBC(fossil).*

**Response:** We thank the reviewer pointing this out. It should be EC not eBC. We apologize for this mistake. The correction has been made:

[Figure]

**FigureR4.** Linear regression of the daily averaged $NO_x$ versus the daily averaged elemental carbon (EC) emitted from diesel vehicular emissions. The red line is the linear fit.

*Figure S8: The caption is not very informative. Do L24h and S24h correspond to L(bj) and R(bj) mentioned in the main text. Being not familiar with the SOM approach, it is hard to figure out what this represents and how to interpret it.*

**Response:** We agree with reviewer. The L24h and S24h correspond to $L_{bj}$ and $S_{bj}$. We have updated the names to make them consistent with the main text. The averaged $L_{bj}$, $S_{bj}$ and $R_{std}$ of the four categories were calculated based on this SOM cluster.

To make this figure easier to interpret, we have added some explanation under the figure as follows:

[Figure]

**FigureR5.** The result of the self-organizing map, sectorial shape represents each variable ($L_{bj}$, $S_{bj}$ and $R_{std}$), the bigger the size the larger this variable. The four circles with the sectorial color together indicate the features of the four motion categories. The bottom left circle means high $L_{bj}$ and $S_{bj}$ but low $R_{std}$, which

indicates the feature of regional scale dominance (RD) category. The bottom right represents the high $R_{std}$ but relatively low $L_{bj}$ and $S_{bj}$ indicating local-scale strong and regional scale weak (LSRW). The upper left represents local scale weak and regional-scale strong (LWRS) and the upper right represent local scale dominance (LD).

**Technical corrections (list non-exhaustive):**

*L32: "It highlights". Do you mean "This study emphasizes the fact that"? (or something along that line)*

**Response:** Yes, we do. We changed the wording:

[revised manuscript text omitted]

*L721: "diurnal" should be replaced by "diel" (24-hour period). At the least, standard deviations could be added for each hourly-averaged point; otherwise each point could be replaced by a box and whiskers that better show the dispersion of the raw data.*

**Response:** The standard deviations have been added for the each point of Figure 2. Now it looks like this:

[Figure]

**Figure R7.** (a) Diel variations of the eBC from fossil fuel combustion (eBC$_{fossil}$) and (b) the eBC from biomass burning (eBC$_{biomass}$), (c) wind speed (m s$^{-1}$) and (d) planetary boundary layer height (m). The black bars of each hourly-averaged point show the standard deviation.

*L724: Figure 3 caption should explain what the different marker colors and the shaded areas represent.*

**Response:** We have revised the figure caption as follows:

[Figure]

**Figure R8.** (a) The 75th – 100th percentile mass concentrations of the eBC from fossil fuel combustion (eBCfossil) and (b) the eBC from biomass burning (eBCbiomass) under local scale dominance (LD, red circle), local scale strong and regional scale weak (LSRW, green circle), local scale weak regional scale strong (LWRS, purple circle) and regional scale dominance (RD, blue circle). $S_{bj}$ is actual wind run distance at 100m height, $R_{bj}$ is the recirculation factor, the grey area indicates good ventilation ($S_{bj} \geq 250km$, $R_{bj} \leq 0.2$), the yellow area indicates air stagnation ($S_{bj} \leq 130km$).

*L731: Figure 5a | The 4th box and whisker plot (blue) should be labelled "fossil" and not "biomass" on the x-axis.*

**Response:** We have re-drawn this figure and corrected the label as follows:

[Figure]

**Figure R9.** Direct radiative effect (DRE) of the eBC from fossil fuel combustion (eBC$_{fossil}$) shaded in grey and the eBC from biomass burning (eBC$_{biomass}$) shaded in yellow (a) in the top atmosphere (TOA), surface (SUF), and the atmosphere atmospheric column (ATM) and (b) the DRE$_{eBC,ATM}$ of two types of eBC under local scale dominance (LD) shaded in light grey labeled as LD, local scale strong and regional scale weak (LSRW) shaded in light blue labeled as LSRW, local scale weak regional scale strong (LWRS) shaded in light grey labeled with LWRS and regional scale dominance (RD) shaded in light blue labelled as RD (c) DRE efficiencies of eBC$_{biomass}$ (shaded in yellow) and eBC$_{fossil}$ (shaded by grey) in TOA, SUF and ATM (d) DRE efficiencies of eBC$_{biomass}$ and eBC$_{fossil}$ at ATM under LD (shaded in light grey labeled as LD), LSRW (shaded in light blue labeled as LSRW), LWRS (shaded in light grey labeled as LWRS) and RD (shaded in light blue labeled with RD).

**Supplementary Information**

*Table S2 would look better if numbers were aligned with column titles.*

**Response:** Thanks for the suggestion, we have aligned the number with column titles.

**Table R7.** Mass absorption cross sections (MAC) and absorption Ångström exponents (AAE) derived from the positive matrix factorization model

|  | Diesel vehicular missions | Coal combustion | Biomass burning | Fossil fuel combustion |
|---|---|---|---|---|
| MAC (m$^2$ g$^{-1}$) | 6.7 | 7.5 | 9.5 | 7.1 |
| AAE | 1.07 | 1.74 | 2.13 | 1.26 |

Figure S1: Add a scale on both maps. Topography should be indicated on the closer range map.

**Response:** We have updated the map of the region in the supplementary materials. It can be seen in the response above.

*Figure S2 caption: "the modelled b$_{abs}$(500 nm) (respectively, SSA) and the observed b$_{abs}$(520 nm) (resp. SSA)". Right now, it reads like a ratio of the two parameters.*

**Response:** We thank reviewer's suggestion and have rewritten the caption as follows:

[Figure]

**Figure R10.** The difference between the modeled $b_{abs}$(500nm) (respectively, SSA) and the observed $b_{abs}$(520nm) (respectively, SSA).

*Figures S4, S5 and S6: What does each point represent? (daily average, other?)*

**Response:** The $b_{abs}(\lambda)$ in Figure S4 are daily averaged data. $NO_x$ and EC in Figure S5 are daily averaged data. The $eBC_{biomass}$ and $eBC_{fossil}$ are hourly averaged data. We have added this into the caption for each figure as follows:

[Figure]

**Figure R11.** Relationship between the observed daily averaged $b_{abs}(\lambda)$ and positive matrix factorization modeled $b_{abs}$ ($\lambda$). $\lambda$ includes 370, 470, 520, 590, 660 and 880nm. The red line is a linear fit.

[Figure]

**Figure R12.** Linear regression of the daily averaged $NO_x$ versus the daily averaged elemental carbon (EC) emitted from diesel vehicular emissions. The red line is the linear fit.

[Figure]

**Figure R13.** Relationship between hourly averaged eBC from biomass burning (eBC$_{biomass}$) and the eBC emitted from fossil fuel combustion (eBC$_{fossil}$). The red line is a linear fit.

*Figure S9: "diurnal" should be replaced by "diel" (24-hour period). At the least, standard deviations could be added for each hourly-averaged point; otherwise each point could be replaced by a box and whisker that better shows the dispersion of the raw data.*

**Response:** We have revised this figure with standard deviations:

[Figure]

**Figure R14.** Diel variations of planetary boundary layer height (PBLH, m) under the dominance of regional scale of motion (RD), the blue dots represent the hourly-averaged PBLH and the black lines represent the standard deviations for each point.

---

## Author Response (AR2)

Comments to the author:

I would like to thank the authors for addressing the reviewers' comments. I particularly appreciate the efforts to improve text clarity and the English grammar and style. I would suggest to further revise the grammar, because I have the impression that during the corrections some typos have been introduced. Here is a list of a few examples (non-exhaustive):

Line 730: "of light-emitted diods" is repeated twice
Line 742: The samples instead of those samples
Line 744: "the ions, which were determined…"
Line 751: "with a test standard sample" or " with test standard samples
Line 874 : "pyrolytic carbon (PC)" instead of "pyrolyzed carbon (OP) produced"
Line 878: "indicate the ratio for which secondary OC and eBC are independent"
Line 880: "Concentration of NOx, wind speed, and direction …"
Line 939: "dust and secondary aerosol"
Line2121: "which is another river valley…" (remove "a")

**Response:** We thank the editor's suggestion, and we have revised the manuscript accordingly.